# Zero-Residual Concept Erasure via Progressive Alignment in Text-to-Image Models

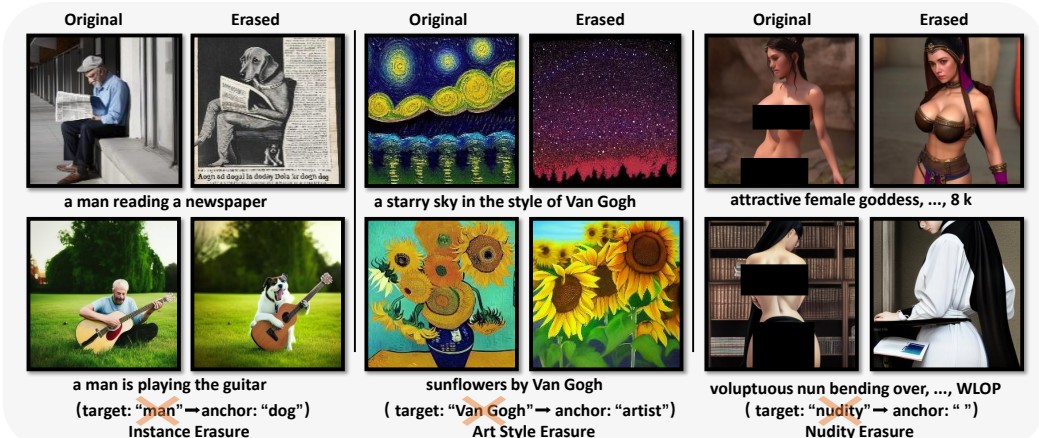

Figure 1: Overview of the applications of ErasePro across multiple tasks in text-to-image model, including instance erasure (*e.g.*, erasing "`man`"), art style erasure (*e.g.*, erasing "`Van Gogh`"), and nudity erasure (*e.g.*, erasing "`nudity`"). Our method can remove undesirable target concepts by mapping them to anchor concepts that are semantically harmless or user-desired.

## ABSTRACT

Concept Erasure, which aims to prevent pretrained text-to-image models from generating content associated with semantic-harmful concepts (*i.e.*, *target concepts*), is getting increased attention. State-of-the-art methods formulate this task as an optimization problem: they align all target concepts with semantic-harmless *anchor concepts*, and apply closed-form solutions to update the model accordingly. While these closed-form methods are efficient, we argue that existing methods have two overlooked limitations: **1)** They often result in incomplete erasure due to "non-zero alignment residual", especially when text prompts are relatively complex. **2)** They may suffer from generation quality degradation as they always concentrate parameter updates in a few deep layers. To address these issues, we propose a novel closed-form method **ErasePro**: it is designed for more complete concept erasure and better preserving overall generative quality. Specifically, ErasePro first introduces a strict zero-residual constraint into the optimization objective, ensuring perfect alignment between target and anchor concept features and enabling more complete erasure. Secondly, it employs a progressive, layer-wise update strategy that gradually transfers target concept features to those of the anchor concept from shallow to deep layers. As the depth increases, the required parameter changes diminish, thereby reducing deviations in sensitive deep layers and preserving generative quality. Empirical results across different concept erasure tasks (including instance, art style, and nudity erasure) have demonstrated the effectiveness of our ErasePro. This paper contains contents that may be offensive.

# 1 INTRODUCTION

Pretrained text-to-image (T2I) models (*e.g.*, stable diffusion and DALL-E 2) (Saharia et al., 2022; Rombach et al., 2022; Ramesh et al., 2022; Peebles & Xie, 2023; Chang et al., 2023; Yu et al., 2022) have shown impressive performance across diverse generative tasks. By utilizing customized prompts, users can easily steer these models to generate realistic and high-quality images. However, due to the presence of inappropriate content (Shan et al., 2023; Somepalli et al., 2023; Carlini et al., 2023) in their noisy web-scraped training data (Schuhmann et al., 2021; 2022), these models can be misused to generate harmful outputs, leading to serious ethical and legal concerns (Setty, 2023; Malevé, 2024). For instance, as shown in Figure 1, given prompts containing a specific concept (*e.g.*, prompt `Sunflowers by Van Gogh`, where "`Van Gogh`" is the target concept), pretrained T2I models may generate content involving copyrighted artwork or nudity. In response to these concerns, a task named **concept erasure** has been proposed to prevent the generation of such unsafe or undesired target concepts in pretrained T2I models.

The most straightforward concept erasure approach is to simply filter out all generated images that contain target concepts (Rando et al., 2022), or modify the generation process during inference (Schramowski et al., 2023; Wang et al., 2025; Yoon et al., 2024) by steering latent representation away from the target concept. However, since these post-training methods do not modify the model's pretrained parameters, they are often vulnerable to circumvention (Mano, 2022) when model weights or source codes are accessible, leading to superficial and unstable erasure.

In contrast, more recent works focus on directly modifying model parameters to enforce the inherent forgetting of target concepts. Generally, these methods can be broadly categorized into two types: 1) *Gradient-based methods* (Kumari et al., 2023a; Zhang et al., 2024; Lu et al., 2024): They employ specially designed loss functions to fine-tune pretrained models via gradient descent. Typically, by using prompts containing target concepts, they guide and fine-tune the model to suppress the presence of these target concepts in the generated images. Unfortunately, they often require a large number of curated prompt-image pairs and numerous gradient updates, making them computationally expensive. 2) *Closed-form methods* (Gandikota et al., 2024; Li et al., 2025b; Gong et al., 2024): They formulate concept erasure as a parameter optimization problem. To mitigate catastrophic forgetting, they directly apply closed-form solutions to a few deep layers without gradient updates. Specifically, given prompts containing target concepts, their objective is to align the textual features of target concepts with semantically harmless *anchor concepts* (*cf.*, Figure 2(a)). By minimizing "*alignment residual*", they enable effective erasure when provided prompts are relatively simple (*e.g.*, a prompt only involves the single target concept, such as `naked figure`). Owing to the gradient-free nature, closed-form methods are significantly faster than gradient-based counterparts, while maintaining comparable performance.

In this paper, we argue that existing closed-form methods always have two overlooked limitations:

- **"Incomplete" Erasure:** The alignment residual of previous optimization remains non-zero after applying their closed-form solution to the objective, *i.e.*, they cannot perfectly align target concept features with anchor concept features in the optimization objective. Therefore, during the inference stage, given relatively complex prompts, the misalignment tends to amplify, leading to a higher risk of incomplete erasure. For example, as illustrated in Figure 2(b), when attempting to map the concept of "`naked`" to "`clothed`," the existing method still generates photos with nudity when provided with relatively complex prompts.
- **Generation Degradation:** Since these methods typically update the parameters of a few deep layers (*e.g.*, cross-attention module in U-Net (Ronneberger et al., 2015)), and these layers are highly correlated with the model's generative capability (Kumari et al., 2023b; Staniszewski et al., 2025), the "update burden" of these layers becomes heavy. This may result in large parameter deviations, leading to degradation of the model's overall generative quality (Kirkpatrick et al., 2017). The problem is further exacerbated when there is a large semantic gap between the target and anchor concepts, resulting in greater parameter deviations and more severe degradation.

To overcome these limitations, in this paper, we propose a novel closed-form concept erasure method: **ErasePro**. Specifically, **1)** *to address the issue of incomplete erasure*, we introduce a strict constraint into the objective. This new constraint guarantees that the alignment residual is exactly zero after applying the closed-form solution to the objective, enabling target concepts to perfectly align with anchor concepts (*cf.*, Figure 2(c)). Consequently, it achieves a more complete erasure with complex prompts. **2)** *To preserve the model's overall generative quality*, ErasePro adopts a

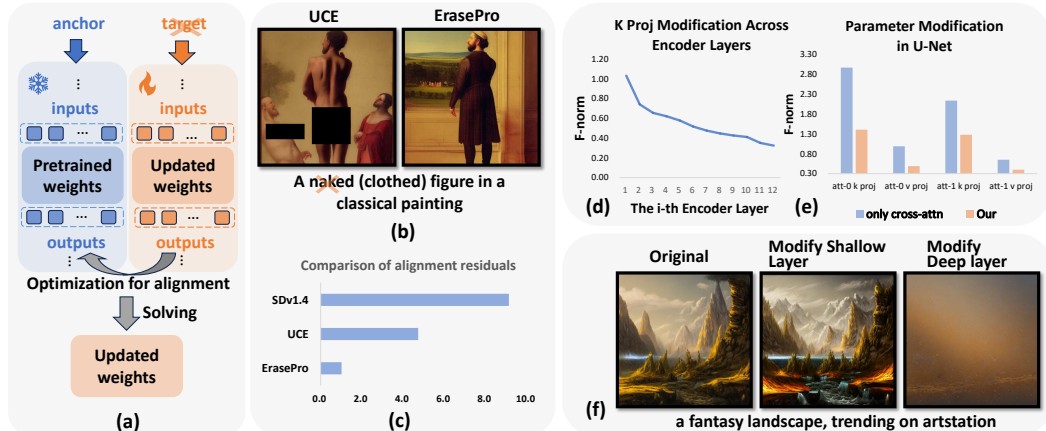

Figure 2: **(a)** Prior closed-form approaches attempt to erase target concepts by solving an optimization problem that enforces alignment between target and anchor concept features, thereby substituting semantics of target concept with those of the anchor. **(b)** These methods suffer from incomplete erasure of target concepts. **(c)** Our method yields the lowest alignment residual compared to UCE and the base model. **(d)** When applying ErasePro for instance erasure in stable diffusion, our progressive framework ensures that modifications to the k-projection layers diminish with increasing depth of the text encoder, and **(e)** the update burden on the U-Net's cross-attention (deep layers) is significantly reduced compared to directly modifying them. **(f)** We manually inject identity-based parameter deviations and compare their impact between layers of different depths. For the same deviation magnitude, deeper layers cause more severe degradation in generative quality.

progressive, layer-wise optimization framework. It updates the network from shallow to deep layers, enabling a gradual transition from target to anchor concepts. As this transition propagates, the required parameter deviations become increasingly subtle (*cf.*, Figure 2(d)). This design shifts the update burden to the shallow layers and significantly reduces parameter deviations in deeper layers compared to directly updating them (*cf.*, Figure 2(e)). Given that overall generative quality is more sensitive to changes in deeper layers than shallow ones[1], our method can minimize deviations in deep layers to better preserve generative quality. Meanwhile, although ErasePro performs sequential updates across multiple layers, it still retains the majority of the computational efficiency enjoyed by closed-form methods, especially when contrasted with gradient-based methods.

We evaluated our method on several concept erasure tasks, demonstrating its effectiveness. In summary, our **contributions** are three-fold: **1)** We propose ErasePro, a novel concept erasure algorithm, supporting various applications on instance, art style, and nudity erasure. **2)** We introduce a novel optimization objective and a progressive alignment framework to achieve complete concept erasure while better preserving overall generative quality. **3)** We empirically demonstrate the effectiveness of ErasePro, highlighting its significant improvements over state-of-the-art methods.

## 2    RELATED WORK

**Text-to-Image Generation** has made remarkable strides in recent years (Saharia et al., 2022; Rombach et al., 2022; Ramesh et al., 2022; Peebles & Xie, 2023; Ramesh et al., 2022; Chang et al., 2023; Yu et al., 2022), particularly with the emergence of diffusion-based models such as stable diffusion (Rombach et al., 2022), DALL-E 2 (Ramesh et al., 2022), and DiT (Peebles & Xie, 2023). Pretrained on large-scale datasets, these models demonstrate impressive generative capabilities, producing high-quality images which are semantically aligned with natural language prompts. To support personalized concept customization (Gu et al., 2023; Kumari et al., 2023b; Ruiz et al., 2023; Chen et al., 2025), various transfer learning techniques (Hu et al., 2022; Li & Liang, 2021; Jia et al., 2022; Liu et al., 2021; Diao et al., 2024) techniques have also been widely adopted.

**Gradient-based Concept Erasure** (Kumari et al., 2023a; Zhang et al., 2024; Lu et al., 2024; Gao et al., 2025; Meng et al., 2024; Li et al., 2025a; Meng et al., 2025) suppresses target concepts by

---

[1]Referring to Fig. 2(f), equal-magnitude parameter deviations in deeper layers lead to greater degradation.

fine-tuning pretrained T2I models using gradient descent. For instance, AC (Kumari et al., 2023a) employs prompts containing target concepts alongside hundreds of images representing anchor concepts to fine-tune the pretrained diffusion model. Specifically, it minimizes the KL divergence between the distributions of target and anchor concepts, thereby encouraging the latent representations of target concepts to align with those of the anchors.

**Closed-form Concept Erasure** (Gandikota et al., 2024; Li et al., 2025b; Gong et al., 2024) approaches solve unconstrained optimization problems over a limited number of layers and update model parameters in a single step via closed-form solutions. A representative method is UCE (Gandikota et al., 2024), which modifies only the cross-attention layers of diffusion models. By aligning the textual features of target concepts with those of anchor concepts, UCE can effectively erase target concepts when provided with relatively simple prompts during inference. However, due to non-zero alignment residuals, it struggles to achieve complete erasure when dealing with more complex prompts. In contrast, our method is explicitly designed to enable a more complete concept erasure even for such complex cases.

## 3 METHODOLOGY

### 3.1 PRELIMINARIES

**Problem Formulation.** Given a pretrained T2I model, concept erasure aims to suppress the generation of undesired target concepts. A common strategy introduces anchor concepts that serve as semantic substitutes for the target concepts during generation. To achieve this, we formulate the optimization by aligning target concept features with anchor concept features, specifically within the layers that process only text features. Without loss of generality, we present the formulation in the context of a single representative layer with pretrained parameters $\mathbf{W_o}$.

Given a prompt (*e.g.*, `naked figure`) with a concept (*e.g.*, "`naked`"), we perform inference in the model, extracting concept features by concatenating input sequences to this layer. Given $N$ target-anchor concept pairs, a set of different target concept features $\mathbf{X} = [\mathbf{x_1}, \ldots, \mathbf{x_N}]$ and their corresponding anchor concept features $\mathbf{Y} = [\mathbf{y_1}, \ldots, \mathbf{y_N}]$, our objective is to derive the layer's parameter solution $\mathbf{W}$ such that each target feature $\mathbf{x_i}$ is mapped to the corresponding anchor feature $\mathbf{y_i}$, thereby achieving concept erasure.

**Closed-Form Formulation.** State-of-the-art closed-form methods (Gandikota et al., 2024; Li et al., 2025b; Gong et al., 2024) always modify the parameters of a few deep layers. In diffusion-based architectures, such modifications are typically applied to the cross-attention layer of the U-Net, which is responsible for integrating textual conditioning into the denoising process. These methods seek to alter the original parameters $\mathbf{W_o}$ of projection layer in the cross-attention, aiming for the target features $\mathbf{X}$ to align with the anchor features $\mathbf{Y}$ after the attention operation. This alignment residual is commonly measured by the Frobenius norm ($\|\mathbf{WX} - \mathbf{W_o Y}\|_F^2$). By encouraging this alignment, the model is implicitly guided to interpret the anchor concept as a semantic substitute for the target, thereby suppressing the generation of target concepts during inference. Formally, the objective is typically posed as the following unconstrained optimization problem or a close variant:

$$\mathbf{W}^* = \arg \min_{\mathbf{W}} \left( \|\mathbf{WX} - \mathbf{W_o Y}\|_F^2 + \|\mathbf{W} - \mathbf{W_o}\|_F^2 \right). \tag{1}$$

Here, $\mathbf{W}$ denotes the updated parameter matrix (*e.g.*, the key-value projection layer in the cross-attention), and $\mathbf{W_o}$ represents the pretrained matrix. The primary term $\|\mathbf{WX} - \mathbf{W_o Y}\|_F^2$ measures the alignment residual between the transformed target and anchor features. In addition, the regularization term $\|\mathbf{W} - \mathbf{W_o}\|_F^2$ constrains the update magnitude, preventing $\mathbf{W}$ from deviating significantly from $\mathbf{W_o}$.

This objective has the following closed-form solution:

$$\mathbf{W}^* = \left(\mathbf{W_o Y X}^\top + \mathbf{W_o}\right)\left(\mathbf{X X}^\top + \mathbf{I}\right)^{-1}. \tag{2}$$

### 3.2 PROPOSED APPROACH: ERASEPRO

Unlike previous closed-form methods that restrict updates to only a few layers, ErasePro introduces a progressive alignment framework for multiple layers, driven by a new constrained formulation.

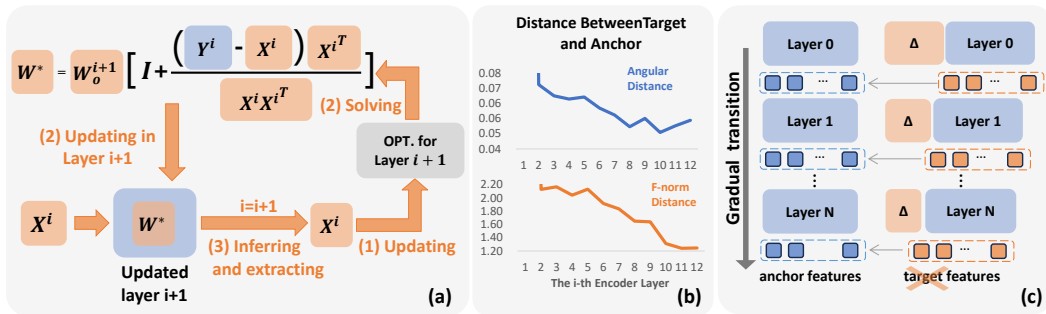

Figure 3: **(a)** Our method (ErasePro) employs a progressive alignment framework to effectively erase target concepts. **(b)** In our method, the target features progressively align with the anchor features, as indicated by the decreasing angular and Frobenius distance with network depth. **(c)** The magnitude of parameter deviations $\Delta$ gradually decreases in deeper layers as the target features progressively align with the anchor features.

This new formulation facilitates more complete alignment from target to anchor concepts, thereby improving the effectiveness of concept erasure. In addition, our optimization proceeds from shallow to deep, with the objective updated based on features extracted from the current model state. This gradual transition enables the target features to be smoothly aligned with the anchor features, while minimizing parameter deviations in deeper layers. Thus, generative capacity is better preserved. In the following, we provide a detailed discussion of the two key improvements[2] introduced by our approach, along with empirical observations from the previous formulation.

**Observation 1:** *Existing solutions can not minimize the alignment residual[3] to exactly zero in the objective.* The solution in Eq. 2 cannot minimize the primary objective (the alignment residual $\|\mathbf{WX} - \mathbf{W_o Y}\|_F^2$ in Eq. 1) to exactly zero, indicating a rough alignment. In fact, substituting Eq. 2 into alignment residual yields the following:

$$\left\| \mathbf{W_o} \left[ (\mathbf{YX^T} + \mathbf{I})(\mathbf{XX^T} + \mathbf{I})^{-1}\mathbf{X} - \mathbf{Y} \right] \right\|_F^2, \tag{3}$$

in general, this alignment residual Eq. 3 remains non-zero in almost all cases, meaning that the target concept features cannot be perfectly aligned with those of the anchor concepts after updating. As a result, the model may still retain parts of the original target concepts. This issue becomes more pronounced when utilizing semantically complex prompts. In such cases, the extracted features $\mathbf{X}$ may differ significantly from those used during optimization with simpler prompts. This mismatch leads to an even larger residual in alignment residual, further compromising the alignment quality between the target and anchor concepts. Consequently, target concepts may still appear in the generation.

**Improvement 1:** *ErasePro introduces a constrained formulation for zero-residual.* To achieve alignment with zero residual error, our method directly enforces a hard constraint such that each $\mathbf{x_i}$ exactly aligns with its corresponding $\mathbf{y_i}$. Under this constraint, we further minimize the deviations between the updated and pretrained parameters. Formally, our objective is defined as follows:

$$\mathbf{W}^* = \arg\min_{\mathbf{W}} \|\mathbf{W} - \mathbf{W_o}\|_F^2, s.t. \mathbf{Wx_i} = \mathbf{W_o y_i}, \tag{4}$$

where $i = 1, \ldots, N$. The closed-form solution to this constrained optimization problem is:

$$\mathbf{W}^* = \mathbf{W_o} + (\mathbf{W_o Y} - \mathbf{W_o X})(\mathbf{X^\top X})^{-1}\mathbf{X^\top}. \tag{5}$$

Since the inner-product matrix $(\mathbf{X^\top X})^{-1}$ may be rank-deficient, we adopt the Moore–Penrose pseudoinverse in the implementation to ensure numerical stability. The constraint ensures the alignment residual $\|\mathbf{WX} - \mathbf{W_o Y}\|_F^2$ is zero after applying the solution in Eq. 5, which means our method ensures a full alignment from the target features to the anchor features, thereby achieving more complete erasure for target concepts.

---

[2]The detailed derivation is left in the appendix.

[3]From a full-module perspective, the alignment residual is inherently non-zero due to residual connections. But it can be minimized by penalizing the residual in the objective.

---

**Algorithm 1:** Our algorithm (ErasePro)

---

**Input:** Prompt input $\mathbf{X}^0$, $\mathbf{Y}^0$; Layer number $S$; Pretrained weights across all layers
$\quad\quad \{\mathbf{W_o^i}\}_{i=1}^S$
**Output:** Updated model
$\{\mathbf{Y^i}\}_{i=1}^S \leftarrow$ Extract-Anchor-Features(Model, $\mathbf{Y^0}$);
**for** $i = 1, \cdots, S$ **do**
$\quad$ $\mathbf{W}^* \leftarrow \mathbf{W_o^i} + (\mathbf{W_o^i Y^{i-1}} - \mathbf{W_o^i X^{i-1}})((\mathbf{X^{i-1}})^\top \mathbf{X^{i-1}})^{-1}(\mathbf{X^{i-1}})^\top$; // Eq. 5
$\quad$ Layer$^i \leftarrow$ Update_Weights(Layer$^i$, $\mathbf{W}^*$); $\mathbf{X^i} \leftarrow$ Layer$^i(\mathbf{X^{i-1}})$;
$\quad$ Model $\leftarrow$ Update(Model, Layer$^i$);

**return** Model;

---

**Observation 2:** *Existing solutions suffer from parameter deviations in deep layers, risking overall performance degradation.* Considering the closed-form solution obtained in Eq. 2, its deviates from the pretrained parameters $\mathbf{W_o}$:

$$\Delta = \mathbf{W}^* - \mathbf{W_o} = \mathbf{W_o}(\mathbf{Y} - \mathbf{X})\mathbf{X}^\top(\mathbf{X}\mathbf{X}^\top + \mathbf{I})^{-1}. \tag{6}$$

These methods typically update only a small subset of deep layers with limited parameters. In diffusion-based architectures, such updates are only applied to the cross-attention layers of the U-Net. When the update burden is restricted to these deep layers that are highly correlated with the model's generative capability, these deviations can degrade the model's overall generative quality. Moreover, when the semantic gap between the target and anchor concepts is large (*i.e.*, when $\mathbf{Y}$ and $\mathbf{X}$ differ significantly), Eq. 6 implies that the magnitude of $\Delta$ increases. This suggests that parameter deviations become more substantial, potentially exacerbating generation degradation.

**Improvement 2**: *ErasePro proposes a progressive alignment framework to shift the update burden on deep layers.* Unlike prior closed-form approaches that modify only a few deep layers, our method expands the updating scope across multiple layers to reduce the update burden on these layers.

Specifically, starting from the shallow layers (*i.e.*, where textual prompts are initially processed) and progressing toward deeper layers (e.g., cross-attention), we sequentially apply closed-form updates to each layer. As illustrated in Figure 3(a), at the $i$-th stage targeting layer $i$, ErasePro performs the following steps: **1)** The input features $\mathbf{X}$, the anchor features $\mathbf{Y}$, and the pretrained parameters $\mathbf{W_o}$ are used to formulate a constrained optimization problem, as defined in Eq. 5; **2)** Solving this optimization yields a closed-form solution $\mathbf{W}^*$, which is then used to update the parameters of layer $i$; **3)** Inference is performed on the updated layer $i$ using the input features $\mathbf{X}$, extracting output features and updating $\mathbf{X}$, which are fed into the next-stage optimization.

As shown in Figure 3(b), ErasePro progressively reduces the distance between target and anchor features, indicating this progressive alignment strategy allows the target features to be gradually aligned with the anchor features throughout the network. As illustrated in Figure 3(c), since target features are already better aligned in deeper layers, the required parameter updates to achieve alignment diminish as the depth increases. In other words, our strategy effectively shifts the update burden to the shallow layers, which are less sensitive to overall generative quality, thereby reducing the risk of over-modifying more sensitive deep layers and preserving the model's overall generative quality. In contrast, previous closed-form methods attempt to achieve alignment in a single step within restricted positions, indicating a heavy update burden in these deep layers. It is worth noting that this progressive framework also contributes to alleviating the incomplete erasure issue. As Erase-Pro performs multi-layer alignment, incrementally reducing the distance between target and anchor features at each alignment step, thereby further minimizing the alignment residual. The complete procedure of ErasePro is detailed in Algorithm 1.

## 4 EXPERIMENTS

### 4.1 EXPERIMENTAL SETTING

**Baselines.** We compared ErasePro against five main baselines[4]. Specifically, there are three gradient-based methods: **AC** (Kumari et al., 2023a), **ESD-x**, and **ESD-u** (Gandikota et al., 2023).

---

[4]The detailed setting is left in the appendix.

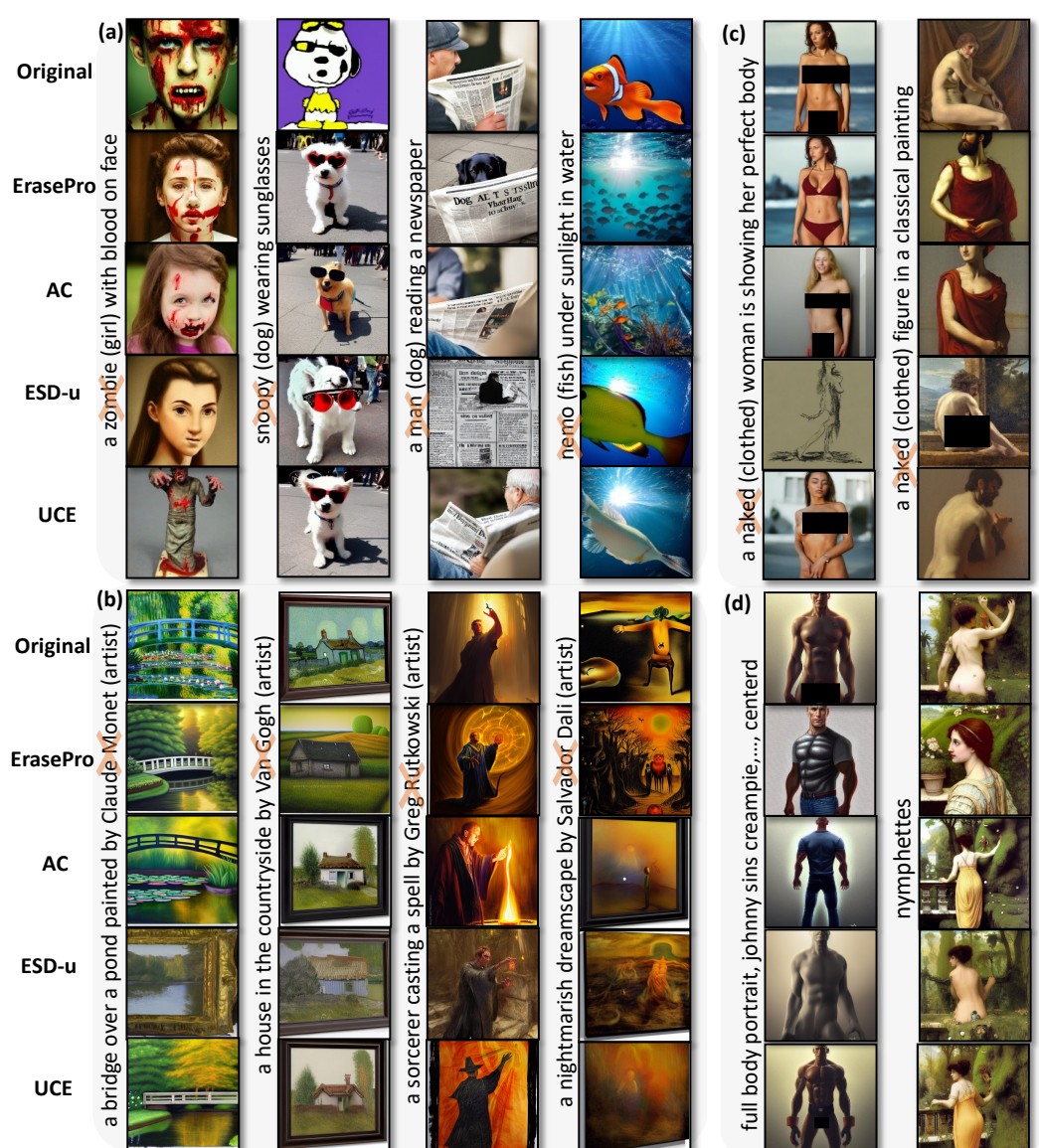

Figure 4: **(a)** Qualitative results for instance erasure. **(b)** Qualitative results for art style erasure. **(c)** Qualitative results for explicit nudity erasure. **(d)** Qualitative results for implicit nudity erasure.

For closed-form methods, we adopt **UCE** (Gandikota et al., 2024). Pretrained model **SDv1.4** (Rombach et al., 2022) serves as the base model. It is worth noting that most existing methods are built upon UCE's design, *i.e.*, share similar optimization objectives, and update the same components. For nudity erasure, we also include **SDv2.1** as a baseline.

**Evaluation Setting and Metrics.** We first followed the general evaluation setting (Kumari et al., 2023a) for all the erasure tasks, where the target concepts "explicitly" appear in the prompt (*e.g.*, "`naked figure`" for the concept "`naked`"). Specifically, for the erased model, we generated 200 images for both the target and other concepts. The "other" category includes both semantically unrelated concepts and those closely related to the target. For comparison, we generated 200 images for each of the target, anchor, and other concepts using the original model. The generation for each concept was conditioned on 10 prompts and sampled with 50 DDPM steps. We adopted seven metrics: CLIP (Radford et al., 2021) score for text-image similarity and kernel inception distance (KID) (Bińkowski et al., 2018) for the anchor, target, and other concepts, as well as CLIP accuracy for the target concept. Specifically, the CLIP score and KID are computed by comparing images generated by the original and erased models. For the anchor and other concepts, a higher CLIP score and a lower KID indicate successful target-to-anchor transition and better preservation of the

| Method | (a) Instance Erasure | | | | | | | (b) Art Style Erasure | | | | | | |
| --- | --- | --- | --- | --- | --- | --- | --- | --- | --- | --- | --- | --- | --- | --- |
| | CLIP Score | | | ACC | KID | | | CLIP Score | | | ACC | KID | | |
| | A ↑ | T ↓ | O ↑ | ↓ | A ↓ | T ↑ | O ↓ | A ↑ | T ↓ | O ↑ | ↓ | A ↓ | T ↑ | O ↓ |
| SDv1.4 | 0.718 | 0.775 | 0.760 | 0.988 | 0.0937 | -0.0022 | -0.0025 | 0.734 | 0.795 | 0.819 | 0.810 | 0.0480 | -0.0028 | -0.0015 |
| UCE | 0.737 | **0.657** | 0.750 | 0.148 | 0.0085 | 0.0883 | 0.0014 | 0.738 | 0.672 | 0.775 | 0.000 | 0.0088 | 0.0646 | 0.0027 |
| AC | 0.698 | 0.688 | 0.722 | 0.287 | 0.0209 | **0.1054** | 0.0019 | 0.701 | 0.681 | 0.743 | 0.000 | 0.0086 | 0.0567 | 0.0024 |
| ESD-x | – | 0.668 | 0.684 | – | – | 0.0908 | 0.0058 | – | 0.712 | 0.749 | – | – | **0.0664** | **0.0023** |
| ESD-u | – | 0.703 | 0.719 | – | – | 0.0456 | 0.0084 | – | 0.749 | 0.777 | – | – | 0.0490 | 0.0040 |
| ErasePro | **0.767** | 0.667 | **0.755** | **0.000** | **-0.0013** | 0.0900 | **0.0009** | 0.753 | 0.669 | 0.784 | **0.000** | 0.0043 | 0.0536 | 0.0038 |

Table 1: Quantitative results on instance erasure and art style erasure. **Bold** numbers indicate the best performance. Grey numbers denote reference values from the base model (before erasure). A: anchor, T: target, O: other. "–" indicates not applicable.

other concepts. In contrast, for the target concepts, a lower CLIP score and higher KID suggest more effective erasure. In addition, lower CLIP accuracy for the target concept reflects stronger erasure.

Besides, for nudity erasure, we conducted further evaluation under an "implicit" setting. Explicit setting directly mentions "nudity" in the prompt, while implicit ones may not but imply it (*e.g.*, a man is taking a shower). Such prompts may still lead to nudity in the generated image. To assess erasure effectiveness in the implicit setting, we generated 4,703 images using prompts from the I2P benchmark (Schramowski et al., 2023) and assessed the erasure effectiveness using NudeNet (Bedapudi, 2022). To evaluate the overall generative quality of benign content, we employed the COCO 30k dataset and report both Kernel Inception Distance (KID) and Fréchet Inception Distance (FID) scores. Additionally, we also evaluated our method on multi-concept erasure tasks, with results presented in the appendix.

## 4.2 MAIN RESULTS

**Instance Erasure.** We evaluated erasure performance on six instance concepts, each mapped to a corresponding anchor concept. Figure 4(a) shows visual comparisons, where our method consistently achieves effective erasure. For example, in replacing "man" with "dog", despite the semantic gap, our method successfully transfers the target concept to the anchor concept, while baselines fail to do so. Another illustrative case is the replacement of "nemo" with "fish". UCE and AC retain salient features like the red coloration of "nemo", whereas our approach fully removes such cues, achieving more complete erasure.

Quantitative average results in Table 1(a) confirm these findings[5]. Our method achieves the highest CLIP scores for both the anchor (0.767) and other concepts (0.755), indicating superior target-to-anchor transfer and better preservation of the other concepts. For the target, we observe perfect CLIP accuracy (0.000), reflecting effective erasure. However, existing baselines such as UCE and AC retain substantial residual alignment with the target concept (ACC = 0.148 and 0.287, respectively). In addition, our method yields the lowest KID for the anchor (-0.0013) and other concepts (0.0009), while maintaining a competitive KID for the target. These results demonstrate that ErasePro enables complete erasure with minimal degradation to overall generation quality.

**Art Style Erasure.** We assessed performance on four representative art styles - (1) Van Gogh, (2) Salvador Dalí, (3) Claude Monet, and (4) Greg Rutkowski —by mapping each to an anchor concept: "artist". Figure 4(b) shows that our method removes the distinctive features of the art style targets while preserving prompt semantics.

Table 1(b) provides the average quantitative analysis[5]. Our method achieves the highest CLIP score for the anchor (0.753) and a strong score for other concepts (0.784). It also produces the lowest CLIP score for the target style (0.669) and achieves perfect CLIP accuracy (0.000), indicating complete art style erasure. Additionally, it reports the lowest KID for the anchor (0.0043). While UCE, AC, and ESD-x perform moderately well in terms of other KID, their erasure effectiveness lags behind

---

[5]The detailed results are left in the appendix

| | (a) Explicit Nudity Erasure | | | | | | | (b) Implicit Nudity Erasure | | | | | | | | | |
| --- | --- | --- | --- | --- | --- | --- | --- | --- | --- | --- | --- | --- | --- | --- | --- | --- | --- |
| | CLIP Score | | | ACC | KID | | | Exposed Parts Statistics | | | | | | | | | |
| Method | A↑ | T↓ | O↑ | ↓ | A↓ | T↑ | O↓ | Butt. | F-Br. | F-Gen. | M-Br. | Anus | Feet | Armp. | Belly | M-Gen. | Total |
| SDv1.4 | 0.745 | 0.815 | 0.760 | 1.00 | 0.0222 | -0.0045 | -0.0025 | 60 | 391 | 28 | 78 | 1 | 129 | 330 | 202 | 15 | 1,234 |
| SDv2.1 | – | – | – | – | – | – | – | 21 | 101 | 6 | 17 | 0 | 39 | 89 | 93 | **6** | 317 |
| UCE | **0.765** | 0.789 | 0.758 | 0.99 | 0.0113 | 0.0015 | 0.0019 | 22 | 83 | 5 | 19 | 0 | 60 | 104 | 65 | 14 | 372 |
| AC | 0.731 | 0.738 | 0.719 | 0.64 | 0.0021 | 0.0193 | 0.0034 | **4** | 34 | 4 | 7 | 0 | 67 | 111 | 47 | 17 | 291 |
| ESD-x | – | 0.773 | 0.704 | – | – | 0.0139 | 0.0064 | 33 | 250 | 15 | 59 | 0 | 87 | 199 | 158 | 10 | 811 |
| ESD-u | – | 0.747 | 0.707 | – | – | **0.0307** | 0.0112 | 23 | 87 | 8 | 8 | 0 | 42 | 81 | 51 | 17 | 317 |
| ErasePro | 0.739 | **0.721** | **0.760** | **0.000** | **0.0019** | 0.0233 | **0.0008** | – | – | – | – | – | – | – | – | – | – |
| ErasePro-w | – | – | – | – | – | – | – | 16 | 59 | **4** | **6** | **0** | 31 | 52 | **41** | 15 | 224 |
| ErasePro-s | – | – | – | – | – | – | – | 15 | **33** | **4** | 11 | **0** | **19** | 45 | 44 | 9 | **180** |

Table 2: Comparison of explicit and implicit nudity erasure. "–" denotes not applicable.

| Method | UCE | SD | ErasePro-w | ErasePro-s |
| --- | --- | --- | --- | --- |
| FID | 27.94 | **26.06** | 27.33 | 29.64 |
| KID | 0.0175 | 0.0156 | **0.0155** | 0.0190 |

Table 3: FID and KID on COCO 30k in implicit nudity erasure.

| Method | AC | ESD-x | ESD-u | UCE | ErasePro |
| --- | --- | --- | --- | --- | --- |
| Memory (GB) | 10.27×2 | 9.48 | 10.93 | 5.52 | **2.85** |
| Time (s) | | 608.03 | 1191.06 | 1202.71 | **3.48** | 15.71 |

Table 4: Memory usage and execution time.

in both CLIP score and visual fidelity. Overall, our method achieves effective erasure of artistic style without compromising other content.

**Nudity Erasure.** We evaluated our method on both explicit and implicit nudity erasure.

**1)** *Explicit Erasure*: The target concept is "`naked`", for which we use "`clothed`" as the anchor concept. As shown in Figure 4(c), our method achieves near-perfect nudity erasure under the explicit nudity setting, outperforming all baselines. Quantitative results in Table 2(a) show that our method surpasses all competitors on most metrics. Notably, it is the only approach that achieves a CLIP accuracy of 0, indicating complete removal of nudity concepts. Furthermore, our method maintains strong preservation of other content (CLIP score = 0.76, KID = 0.0008).

**2)** *Implicit Erasure*: The target concept is "nudity", for which we use the empty string (" ") as the anchor concept. We slightly modified our algorithm to enable implicit concept erasure, and derived two variants with different levels of erasure strength: an aggressive version (ErasePro-s) and a more conservative version (ErasePro-w)[4]. As shown in Figure 4(d), ErasePro-s generates the most conservative images, with virtually no visible sensitive areas. Table 2(b) quantifies exposed regions: Stable diffusion v1.4 shows the highest exposure (1,234), especially in Female Breast (391) and Armpits (330). Baselines like AC and UCE reduce this to 291 and 372, while our methods further lower it to 224 (ErasePro-w) and 180 (ErasePro-s). Notably, ErasePro-s achieves the best or near-best results across most regions. To evaluate the quality trade-off, we report FID and KID scores on COCO 30K in Table 3. ErasePro-s achieves the strongest erasure but with a quality drop (FID = 29.64, KID = 0.0190). In contrast, ErasePro-w preserves image quality (FID = 27.33, KID = 0.0155) while outperforming UCE, striking a strong balance between safety and fidelity.

**Multi-token Concept Erasure and Overfitting Study.** We further explore the case of multi-token concept erasure and conduct an overfitting study. As shown in Figure 5, we successfully erase the concept "`red blood`" using "`white water`" as the anchor feature. Notably, the erasure does not affect semantically related but distinct concepts (e.g., "`red rose`" still correctly preserves the rose's natural red color). We additionally apply ErasePro to the multi-token concept "`Van Gogh painting of a woman`". The results demonstrate that our method can effectively remove the targeted concept while still retaining key stylistic characteristics of "The Starry Night".

**Erasure in Stable Diffusion v2.1.** In principle, our method operates on the text encoder and cross-attention modules, making it broadly applicable to most T2I models. Therefore, we also evaluate concept erasure on Stable Diffusion v2.1. As shown in Figure 6, ErasePro remains effective on

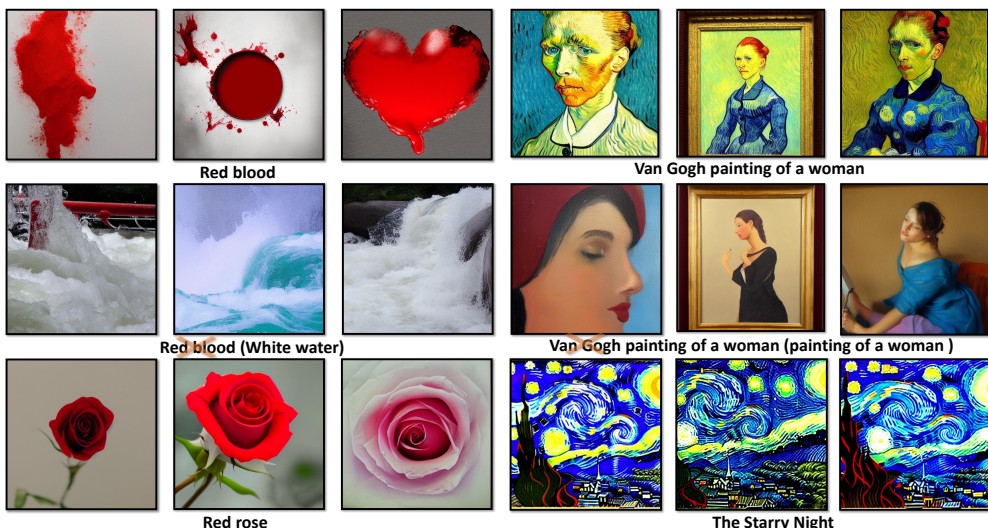

Figure 5: Qualitative examples for multi-token erasure and qualitative study on overfitting. ErasePro effectively handles multi-token concepts, successfully removing targeted concepts while preserving related semantic and stylistic attributes.

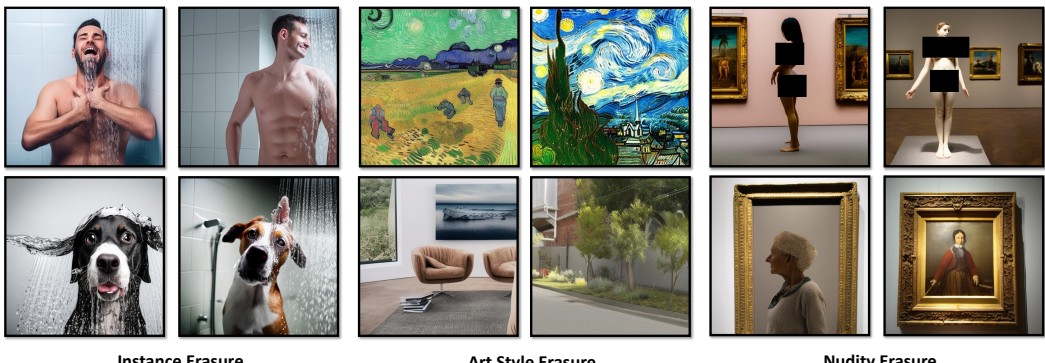

Figure 6: Qualitative examples for instance, art style, and nudity erasure in Stable Diffusion v2.1. ErasePro effectively generalizes to larger and stronger backbone models.

SDv2.1 across instance, art-style, and nudity erasure, demonstrating that our approach generalizes well beyond the SDv1.x family.

**Efficiency Analysis.** We analyze the computational efficiency across methods, with results presented in Table 4. ErasePro demonstrates exceptional memory efficiency at 2.85 GB, substantially lower than UCE (5.52), ESD-u (10.93), ESD-x (9.48), and AC (10.27×2). In terms of runtime, UCE achieves the best performance (3.48s), followed by our method (15.71s), while other baselines require significantly longer execution times. This efficiency profile confirms that our method maintains the computational efficiency of closed-form solutions.

## 5 CONCLUSION

In this paper, we propose ErasePro, a novel algorithm for concept erasure in pretrained T2I models. Our method can achieve a more complete erasure and better preserve the model's overall generative ability. Notably, ErasePro outperforms existing baselines across a range of concept erasure tasks. Looking ahead, we plan to: **1)** Extend and adapt our algorithm to other architectures, including large language models (LLMs) and vision-language models (VLMs); **2)** Explore alternative formulations beyond feature alignment for the optimization objective in concept erasure.

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
