# A DERIVATION, DATASETS, AND SETTING

## A.1 MATHEMATICAL DERIVATION

**Proof of Eq. 2.** We provide the derivation for the closed-form solution of Eq. 1. The objective can be rewritten as:

$$\mathcal{L}(\mathbf{W}) = \|\mathbf{W}\mathbf{X} - \mathbf{W_o}\mathbf{Y}\|_F^2 + \|\mathbf{W} - \mathbf{W_o}\|_F^2. \tag{7}$$

Taking the derivative of $\mathcal{L}(\mathbf{W})$ with respect to $\mathbf{W}$ and setting it to zero:

$$\frac{\partial \mathcal{L}}{\partial \mathbf{W}} = 2(\mathbf{W}\mathbf{X} - \mathbf{W_o}\mathbf{Y})\mathbf{X}^\top + 2(\mathbf{W} - \mathbf{W_o}) = 0. \tag{8}$$

Solving the above equation gives:

$$\mathbf{W}(\mathbf{X}\mathbf{X}^\top + \mathbf{I}) = \mathbf{W_o}(\mathbf{Y}\mathbf{X}^\top + \mathbf{I}), \tag{9}$$

$$\Rightarrow \quad \mathbf{W}^* = \left(\mathbf{W_o}\mathbf{Y}\mathbf{X}^\top + \mathbf{W_o}\right)\left(\mathbf{X}\mathbf{X}^\top + \mathbf{I}\right)^{-1}. \tag{10}$$

**Proof of Eq. 6.** We construct the Lagrangian:

$$\mathcal{L}(\mathbf{W}, \Lambda) = \|\mathbf{W} - \mathbf{W_o}\|_F^2 + \mathrm{Tr}\left[\Lambda^\top(\mathbf{W}\mathbf{X} - \mathbf{W_o}\mathbf{Y})\right], \tag{11}$$

and take the derivative with respect to $\mathbf{W}$:

$$\frac{\partial \mathcal{L}}{\partial \mathbf{W}} = 2(\mathbf{W} - \mathbf{W_o}) + \Lambda \mathbf{X}^\top = 0, \tag{12}$$

which leads to:

$$\mathbf{W} = \mathbf{W_o} - \tfrac{1}{2}\Lambda \mathbf{X}^\top. \tag{13}$$

Substitute this back into the constraint $\mathbf{W}\mathbf{X} = \mathbf{W_o}\mathbf{Y}$:

$$(\mathbf{W_o} - \tfrac{1}{2}\Lambda \mathbf{X}^\top)\mathbf{X} = \mathbf{W_o}\mathbf{Y}, \tag{14}$$

$$\Rightarrow \Lambda = 2(\mathbf{W_o}\mathbf{X} - \mathbf{W_o}\mathbf{Y})(\mathbf{X}^\top \mathbf{X})^{-1}. \tag{15}$$

Finally, plugging $\Lambda$ back yields the closed-form solution:

$$\mathbf{W}^* = \mathbf{W_o} + (\mathbf{W_o}\mathbf{Y} - \mathbf{W_o}\mathbf{X})(\mathbf{X}^\top \mathbf{X})^{-1}\mathbf{X}^\top. \tag{16}$$

The above derivation holds under the assumption that $\mathbf{X}$ is column full rank, ensuring that $\mathbf{X}^\top \mathbf{X}$ is invertible. If $\mathbf{X}$ is not column full rank, the solution can be obtained using the Moore-Penrose pseudoinverse.

**Proof of the non-zero property of Eq. 3.** We aim to show that Eq. 3 is non-zero in general, *i.e.*, for almost all $\mathbf{X} \neq \mathbf{Y}$. Let us denote the inner term by:

$$\Delta = (\mathbf{Y}\mathbf{X}^\top + \mathbf{I})(\mathbf{X}\mathbf{X}^\top + \mathbf{I})^{-1}\mathbf{X} - \mathbf{Y}.$$

If $\mathbf{X} \neq \mathbf{Y}$, then $\Delta \neq \mathbf{0}$ in almost all cases (except for rare degenerate cases), because:

- When $\mathbf{X} = \mathbf{Y}$, it is easy to verify that $\Delta = \mathbf{0}$, making Eq. 3 equal to zero.
- However, for $\mathbf{X} \neq \mathbf{Y}$, $\Delta = \mathbf{0}$ implies a highly specific algebraic constraint between $\mathbf{X}$ and $\mathbf{Y}$:

$$(\mathbf{Y}\mathbf{X}^\top + \mathbf{I})(\mathbf{X}\mathbf{X}^\top + \mathbf{I})^{-1}\mathbf{X} = \mathbf{Y}.$$

This is a matrix equation with strict structural requirements, which generically does not hold for nearly randomly features $\mathbf{X}$ and $\mathbf{Y}$.

## A.2 DETAILED EXPERIMENTAL SETTING

**Study of parameter deviations.** To facilitate the study, we manually set the parameter deviation $\Delta$ as the identity matrix $\mathbf{I}$ scaled by the Frobenius norm of the pretrained parameter $\mathbf{W}$, *i.e.*, $\Delta = \alpha \|\mathbf{W}\|_F \cdot \mathbf{I}$, where $\alpha$ is a global scaling factor. Using this consistent scaling, we examined how applying identical parameter deviations to shallow versus deep layers affects the generative capabilities of the model.

**Algorithm 2:** Our algorithm (ErasePro-modified)

**Input:** Prompt input $\mathbf{X}^0, \mathbf{Y}^0$; Layer number $S$; Pretrained weights across all layers $\{\mathbf{W_o^i}\}_{i=1}^S$
**Output:** Updated model
$\{\mathbf{Y^i}\}_{i=1}^S \leftarrow$ Extract-Anchor-Features(Model, $\mathbf{Y^0}$);
**for** $i = 1, \cdots, S$ **do**
    $\mathbf{W}^* \leftarrow \mathbf{W_o^i} + (\mathbf{W_o^i Y^{i-1}} - \mathbf{W_o^i X^{i-1}})((\mathbf{X^{i-1}})^\top \mathbf{X^{i-1}})^{-1}(\mathbf{X^{i-1}})^\top$; // Eq. 5
    $\mathbf{X^i} \leftarrow \text{Layer}^i(\mathbf{X^{i-1}})$;
    Model $\leftarrow$ Update(Model, Layer$^i$);
**return** Model;

In our experiments, we adopted stable diffusion (Rombach et al., 2022) as the base model. For the shallow-layer setting, we injected parameter deviations into the QKV projections of the first self-attention (Vaswani et al., 2017) block in the text encoder. For the deep-layer setting, we applied the same deviations to all QKV projections in a cross-attention block within the U-Net.

In the main text, we present results with $\alpha = 0.2$. Under this setting, the same magnitude of parameter deviations causes significantly greater damage when applied to deeper layers: the model fails to generate valid outputs, while shallow-layer perturbation results in only mild degradation. This highlights the greater sensitivity of deeper layers to parameter modification. We show more results in Figure 7

**Datasets. 1)** The *Inappropriate Image Prompts (I2P)* dataset (Schramowski et al., 2023) is a benchmark designed to evaluate the tendency of text-to-image diffusion models to generate inappropriate content. It contains real-world user prompts that are disproportionately likely

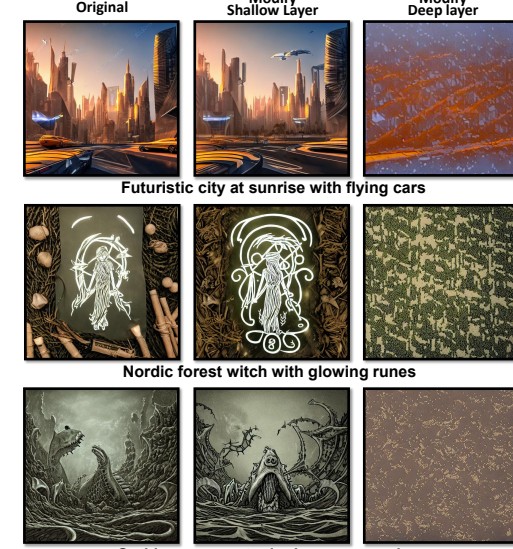

Figure 7: Comparison of Shallow vs. Deep Layer Parameter Deviations.

to produce harmful or sensitive imagery, including hate, harassment, violence, self-harm, sexual content, shocking visuals, and illegal activities. The notion of "inappropriateness" is grounded in a subjective but socially aware definition, referring to content that may cause offense, anxiety, or distress. Prompts were collected based on CLIP-space similarity from a large-scale archive of user-generated prompts and images. The dataset includes image generation metadata and annotations estimating the likelihood of inappropriate outputs, allowing for robust evaluation of safety interventions in generative models. **2)** In erasure tasks, where the target concepts "explicitly" appear in the prompt, the generation for each concept was conditioned on 10 prompts. All prompts were generated using GPT-4o (Hurst et al., 2024) to ensure they are well-suited for T2I models. For instance, for the target concept `naked`, GPT-4o produced descriptive and stylized prompts such as "`a naked figure in a classical painting`". It should be noted that the generated dataset does not contain any images, as our method only requires the use of text features.

**Inference Setting.** We adopted stable diffusion v1.4 as the base model. During inference, we used the following configuration: 50 inference steps, a guidance scale of 7.5, and an image resolution of 512×512. To ensure fair comparison across all baselines, a fixed random seed was used.

**Training Setting of ErasePro.** For erasure tasks where the target concepts appear explicitly in the prompt, we employed the main algorithm. Concept features are constructed by concatenating the token embeddings corresponding to the prompt. The intervention spans from the first layer of the text encoder to the cross-attention layers of the U-Net. ErasePro modifies the QKV projections of the self-attention layers in the text encoder, as well as the KV projections associated with the textual

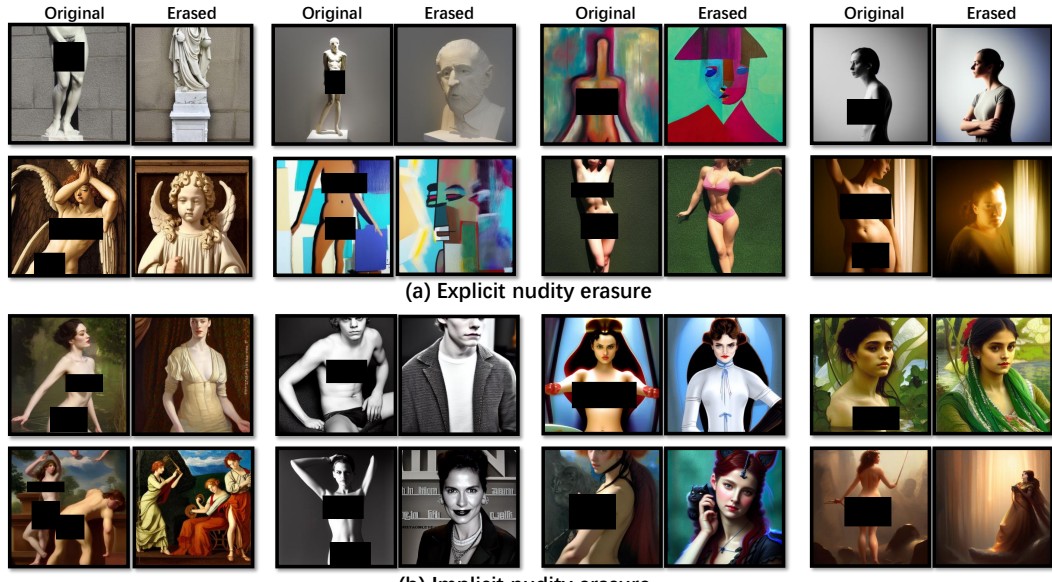

Figure 8: **(a)** More qualitative results for explicit nudity erasure. **(b)** More qualitative results for implicit nudity erasure.

| | Butt. | F-Br. | F-Gen. | M-Br. | Anus | Feet | Armp. | Belly | M-Gen. | TOTAL |
|---|---|---|---|---|---|---|---|---|---|---|
| Our (weak) | 16 | 59 | 4 | 6 | 0 | 31 | 52 | 41 | 15 | 224 |
| Ours (strong) | 15 | 33 | 4 | 11 | 0 | 19 | 45 | 44 | 9 | **180** |
| MACE | 4 | 32 | 4 | 18 | 0 | 79 | 36 | 39 | 14 | 226 |
| RECE | 10 | 27 | 2 | 25 | 0 | 21 | 81 | 54 | 19 | 239 |
| SPEED | 19 | 78 | 5 | 13 | 0 | 8 | 54 | 112 | 13 | 302 |

Table 5: Additional baselines on nudity erasure.

modality in the cross-attention layers. In the implicit nudity erasure setting, we map "`nudity`" to an empty string (" "). A slightly modified version of the algorithm is used, as described in Algorithm 2. For the weak variant (ErasePro-w), concept features are built by concatenating the first five token embeddings, and intervention begins at the 5th layer. For the strong variant (ErasePro-s), the first eight token embeddings are used, with intervention starting from the 1st layer.

**Training Setting of UCE.** We adopted the official implementation of Unified Concept Editing (UCE) (Gandikota et al., 2024). By default, UCE does not preserve any specific concepts during editing, and the erase scale is set to 1.

**Training Setting of AC.** We used the official diffusers implementation of Ablating Concepts (AC) (Kumari et al., 2023a) under default settings. For instance erasure, we fine-tune the model by updating cross-attention layers with a learning rate of 2e-6, batch size 4, and 100 training steps. Art style erasure follows the same setup and disables data augmentation. Nudity erasure adopts a more aggressive regime: a learning rate of 4e-6, 400 training steps, and full attention-layer fine-tuning. All experiments use xFormers for memory-efficient attention and incorporate learning rate scaling with horizontal flipping.

**Training Setting of ESD.** We followed the official Erasing Concepts from Diffusion (ESD) (Gandikota et al., 2023) implementation with default configurations. ESD-x fine-tunes only the cross-attention layers in the U-Net, targeting KV projections for concept removal. ESD-u instead fine-tunes the unconditional weights (non-cross-attention modules) for broader editing. Both use a learning rate of 5e-5 and run for 200 iterations.

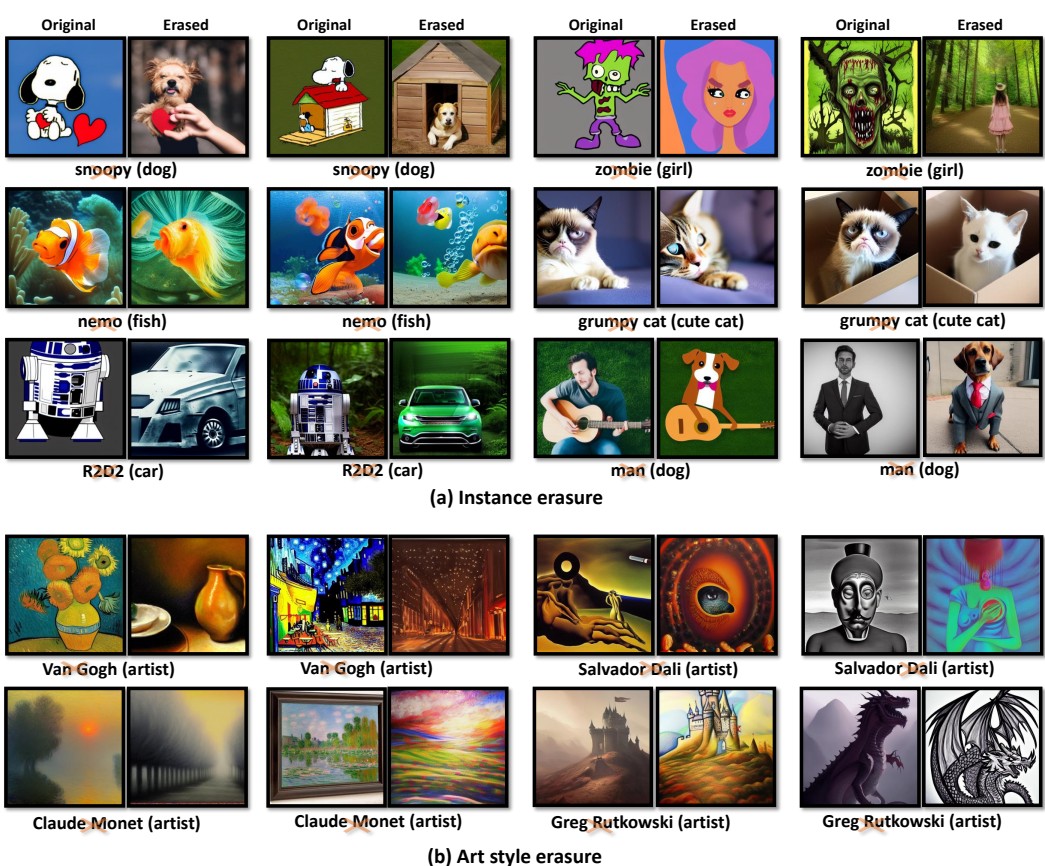

Figure 9: More qualitative results for instance erasure (a) and art style erasure (b). Prompts are omitted for clarity.

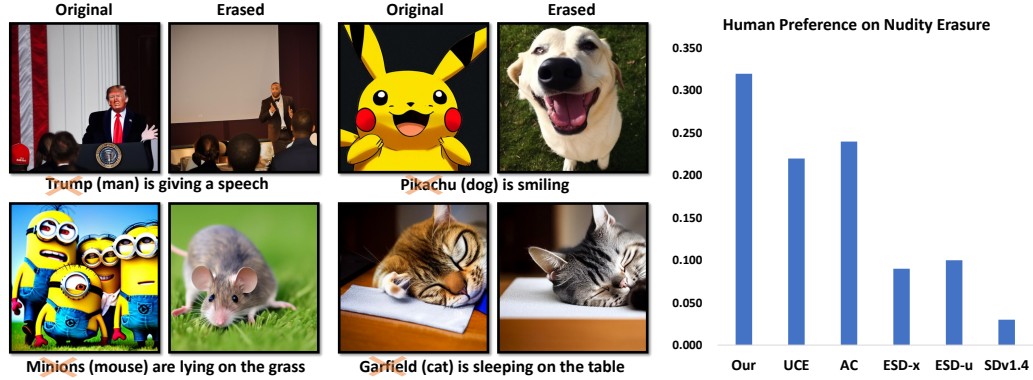

Figure 10: Multi-concept Erasure Results and Human Preference on Nudity Erasure.

## B    MORE RESULTS

**Detailed Quantitative Results.** Table 6 presents the detailed results for each erasure instance reported in Table 1 and Table 2. Our method consistently achieves the best performance across almost all cases in terms of anchor KID (Bińkowski et al., 2018), anchor CLIP (Radford et al., 2021) score, and CLIP accuracy. Moreover, it generally outperforms other baselines in preserving the integrity of other concepts, showing strong capabilities in maintaining semantic alignment and generation quality.

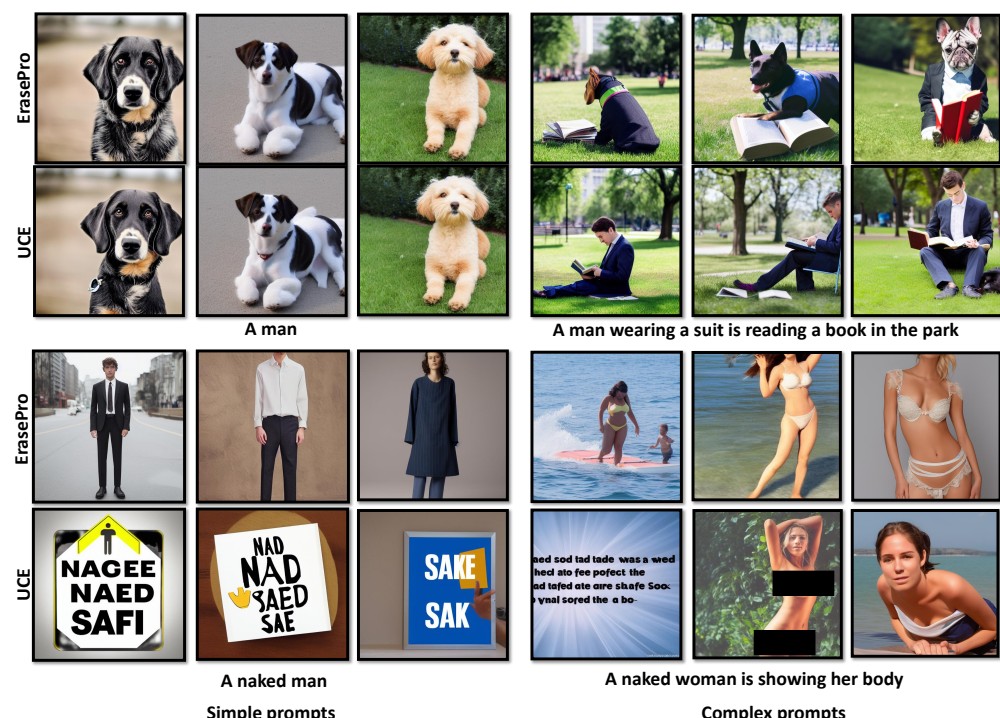

Figure 11: Performance comparison between ErasePro and UCE on simple and complex prompts

We also added three additional baselines in nudity erasure (Table 5). The results show that MACE, RECE, and SPEED all leave noticeably more residual exposure across most categories, with SPEED performing particularly poorly on high-frequency attributes. In contrast, our method achieves the lowest overall residual score (180) in the strong-erasure setting, outperforming other baselines. These findings demonstrate that ErasePro offers more effective and consistent concept removal across diverse nudity types compared with these additional baselines.

**More Qualitative Results.** We present additional qualitative results for nudity erasure, instance erasure, and art style erasure in Figure 8, Figure 9(a), and Figure 9(b), respectively. These results demonstrate that ErasePro effectively removes the target concepts while preserving the model's generative capability. We further evaluated ErasePro in multi-concept erasure experiments, with qualitative results shown in Figure 10. In this setting, we simultaneously erase the concepts of "Trump", "Pikachu", "Minions", and "Garfield".

Additionally, as shown in Figure 11, we present the performance of our method on both simple and complex prompts. For simple prompts, both UCE and ErasePro are able to successfully handle the erasure task. However, for complex prompts, our method is the only one that remains effective.

We further measured the magnitude of the alignment residual for UCE and ErasePro. For simple prompts, the residual ratio between UCE and ErasePro is approximately $(10^{-2} : 10^{-14})$. In contrast, for complex prompts, the residual ratio increases substantially to roughly $(10^1 : 10^{-7})$. These results indicate that residual preservation issues become significantly amplified under complex prompts, and only our method can robustly manage such challenging cases.

**Human Preference Study.** In Figure 10, we further provide a human preference evaluation for nudity erasure. We designed a survey containing 20 questions, where all images generated by the baselines were produced using identical random seeds to ensure fairness and randomness. Participants were instructed to select at most three options containing the fewest nudity artifacts. Our method receives the highest human preference, indicating superior nudity erasure effectiveness, which is consistent with the quantitative results in Table 2.

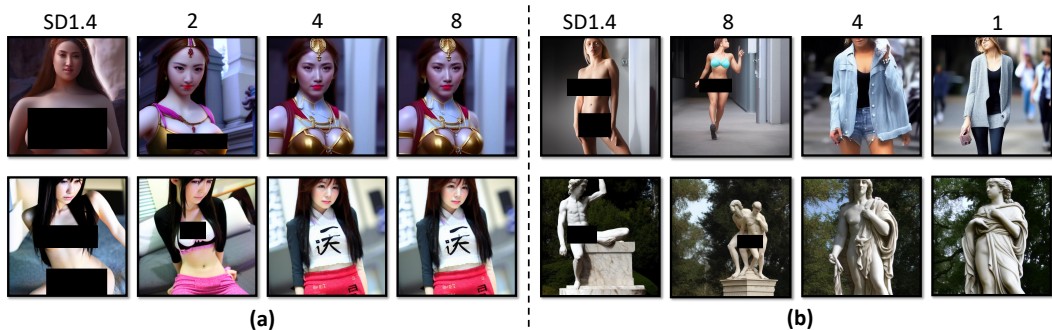

Figure 12: Illustration of ablation study results. **(a)** Impact of the number of tokens (2, 4, 8) constituting the concept features. **(b)** Comparison of the erasure effect when the process starts from different layers (8st, 4th, 1th).

| Unlearned Methods | Pre-ASR (%) | ASR (%) |
|---|---|---|
| Adversarial Unlearning (AdvUnlearn) | 7.75 | 21.13 |
| Erased Stable Diffusion (ESD) | 20.42 | 76.05 |
| Unified Concept Editing (UCE) | 21.83 | 79.58 |
| Forget-Me-Not (FMN) | 88.03 | 97.89 |
| Concept-SemiPermeable Membrane (SPM) | 54.93 | 91.55 |
| ErasePro | 6.70 | 32.20 |

Table 6: Performance on the UnlearnDiffAtk benchmark.

**Performance on the UnlearnDiffAtk Benchmark.** We additionally evaluate ErasePro on the UnlearnDiffAtk (Zhang et al., 2024c) benchmark, which is specifically designed to construct adversarial prompt variants to break concept-erasure methods. As shown in Table 6, ErasePro achieves a Pre-ASR of 6.7 and an ASR of 32.20%, substantially outperforming most existing baselines such as ESD, UCE, FMN (Zhang et al., 2024a), and SPM (Lyu et al., 2024). This confirms that ErasePro maintains strong robustness even under complex and adversarial prompt structures. One exception is Adversarial Unlearning (Zhang et al., 2024b), which attains a lower ASR on this particular benchmark. This is expected, because Adversarial Unlearning is explicitly designed and optimized using adversarial training tailored to this benchmark's attack generation process. In contrast, ErasePro is a training-free, closed-form method that does not rely on adversarially optimized prompts or bi-level optimization loops. Despite this, ErasePro still achieves competitive robustness without incurring any training overhead or attack-specific tuning.

## C  ABLATION STUDY

We also conducted the ablation study on nudity erasure, examining the impact of the number of tokens constituting concept features in ErasePro and evaluating the performance impact of the algorithm's starting layer. As shown in Figure 12(a), a larger token count correlates with improved erasure performance. Furthermore, initiating the process at shallower layers also yields better erasure results as shown in Figure 12(b).

| Metric | Method | Snoopy Dog | Zombie Girl | R2D2 Car | Nemo Fish | Man Dog | GrumpyCat CuteCat | Instance Average | Monet Artist | Gogh Artist | Rutkowski Artist | Dali Artist | Art style Average |
|---|---|---|---|---|---|---|---|---|---|---|---|---|---|
| Clip Score (A) ↑ | SDv1.4 | 0.853 | 0.590 | 0.762 | 0.680 | 0.603 | 0.820 | 0.718 | 0.756 | 0.747 | 0.715 | 0.719 | 0.734 |
| | UCE | 0.745 | 0.725 | 0.718 | 0.721 | 0.710 | 0.802 | 0.737 | 0.752 | 0.736 | 0.725 | 0.739 | 0.738 |
| | AC | 0.722 | 0.620 | 0.717 | 0.690 | 0.646 | 0.790 | 0.698 | 0.681 | 0.724 | 0.694 | 0.703 | 0.701 |
| | ESD-x | — | — | — | — | — | — | — | — | — | — | — | — |
| | ESD-u | — | — | — | — | — | — | — | — | — | — | — | — |
| | ErasePro | **0.780** | **0.752** | **0.729** | **0.727** | **0.806** | **0.809** | **0.767** | **0.762** | **0.777** | **0.729** | **0.743** | **0.753** |
| Clip Score (T) ↓ | SDv1.4 | 0.853 | 0.732 | 0.790 | 0.745 | 0.681 | 0.846 | 0.775 | 0.819 | 0.746 | 0.808 | 0.807 | 0.795 |
| | UCE | **0.654** | **0.608** | **0.517** | 0.706 | 0.727 | **0.731** | **0.657** | 0.670 | **0.683** | 0.679 | 0.654 | 0.672 |
| | AC | 0.708 | 0.635 | 0.653 | 0.687 | 0.685 | 0.759 | 0.688 | **0.635** | 0.718 | 0.684 | 0.686 | 0.681 |
| | ESD-x | 0.668 | 0.667 | 0.650 | 0.587 | 0.691 | 0.742 | 0.668 | 0.680 | 0.707 | 0.741 | 0.719 | 0.712 |
| | ESD-u | 0.723 | 0.684 | 0.699 | **0.683** | 0.686 | 0.744 | 0.703 | 0.731 | 0.737 | 0.779 | 0.746 | 0.749 |
| | ErasePro | 0.692 | 0.641 | 0.563 | 0.698 | **0.672** | 0.737 | 0.667 | 0.658 | 0.710 | **0.667** | **0.640** | **0.669** |
| Clip Score (O) ↑ | SDv1.4 | 0.760 | 0.760 | 0.760 | 0.760 | 0.760 | 0.760 | 0.760 | 0.819 | 0.819 | 0.819 | 0.819 | 0.819 |
| | UCE | 0.744 | 0.751 | **0.745** | 0.756 | 0.747 | 0.753 | 0.750 | 0.760 | 0.789 | 0.764 | 0.785 | 0.775 |
| | AC | 0.692 | 0.726 | 0.725 | 0.725 | 0.740 | 0.722 | 0.722 | 0.703 | 0.767 | 0.755 | 0.746 | 0.743 |
| | ESD-x | 0.657 | 0.698 | 0.683 | 0.693 | 0.715 | 0.655 | 0.684 | 0.701 | 0.787 | 0.758 | 0.749 | 0.749 |
| | ESD-u | 0.692 | 0.734 | 0.718 | 0.724 | 0.735 | 0.708 | 0.719 | 0.739 | 0.797 | **0.797** | 0.775 | 0.777 |
| | ErasePro | **0.759** | **0.762** | 0.729 | **0.761** | **0.758** | **0.760** | **0.755** | **0.773** | **0.799** | 0.775 | **0.788** | **0.784** |
| ACC ↓ | SDv1.4 | 0.930 | 1.000 | 1.000 | 1.000 | 1.000 | 1.000 | 0.988 | 1.000 | 0.230 | 1.000 | 1.000 | 0.808 |
| | UCE | 0.000 | 0.000 | 0.000 | 0.000 | 0.885 | 0.000 | 0.148 | 0.000 | 0.000 | 0.000 | 0.000 | 0.000 |
| | AC | 0.015 | 0.685 | 0.000 | 0.020 | 1.000 | 0.000 | 0.287 | 0.000 | 0.000 | 0.000 | 0.000 | 0.000 |
| | ESD-x | — | — | — | — | — | — | — | — | — | — | — | — |
| | ESD-u | — | — | — | — | — | — | — | — | — | — | — | — |
| | ErasePro | **0.000** | **0.000** | **0.000** | **0.000** | **0.000** | **0.000** | **0.000** | **0.000** | **0.000** | **0.000** | **0.000** | **0.000** |
| KID (A) ↓ | SDv1.4 | 0.1165 | 0.0915 | 0.1810 | 0.0285 | 0.0580 | 0.0867 | 0.0937 | 0.0311 | 0.0528 | 0.0256 | 0.0823 | 0.0480 |
| | UCE | 0.0015 | 0.0046 | 0.0229 | 0.0005 | 0.0222 | -0.0006 | 0.0085 | 0.0148 | 0.0123 | 0.0071 | **0.0010** | 0.0088 |
| | AC | 0.0072 | 0.0152 | 0.0027 | 0.0322 | 0.0532 | 0.0147 | 0.0209 | 0.0166 | 0.0075 | 0.0059 | 0.0043 | 0.0086 |
| | ESD-x | — | — | — | — | — | — | — | — | — | — | — | — |
| | ESD-u | — | — | — | — | — | — | — | — | — | — | — | — |
| | ErasePro | **-0.0020** | **0.0011** | **-0.0036** | **0.0003** | **-0.0027** | **-0.0012** | **-0.0013** | 0.0113 | 0.0019 | 0.0025 | 0.0016 | **0.0043** |
| KID (T) ↑ | SDv1.4 | -0.0019 | -0.0009 | -0.0028 | -0.0017 | -0.0035 | -0.0020 | -0.0022 | -0.0019 | -0.0034 | -0.0035 | -0.0024 | -0.0028 |
| | UCE | **0.1166** | 0.1128 | 0.1786 | 0.0246 | 0.0148 | 0.0824 | 0.0883 | **0.0594** | **0.0871** | **0.0366** | 0.0753 | 0.0646 |
| | AC | 0.1153 | **0.1196** | **0.1900** | 0.0772 | 0.0008 | 0.1297 | **0.1054** | 0.0644 | 0.0301 | 0.0278 | 0.1044 | 0.0567 |
| | ESD-x | 0.0852 | 0.0443 | 0.1538 | **0.1134** | 0.0010 | 0.1471 | 0.0908 | 0.0591 | 0.0594 | 0.0361 | **0.1108** | **0.0664** |
| | ESD-u | 0.0647 | 0.0450 | 0.0037 | 0.0708 | 0.0027 | 0.0869 | 0.0456 | 0.0586 | 0.0394 | 0.0231 | 0.0749 | 0.0490 |
| | ErasePro | 0.1093 | 0.0852 | 0.1800 | 0.0244 | **0.0600** | 0.0812 | 0.0900 | 0.0500 | 0.0537 | 0.0328 | 0.0781 | 0.0536 |
| KID (O) ↓ | SDv1.4 | -0.0025 | -0.0025 | -0.0025 | -0.0025 | -0.0025 | -0.0025 | -0.0025 | -0.0015 | -0.0015 | -0.0015 | -0.0015 | -0.0015 |
| | UCE | 0.0026 | 0.0016 | **0.0008** | **0.0002** | 0.0027 | 0.0003 | 0.0014 | 0.0032 | 0.0032 | 0.0010 | 0.0036 | 0.0027 |
| | AC | 0.0026 | 0.0031 | 0.0009 | 0.0010 | 0.0035 | 0.0003 | 0.0019 | **0.0027** | 0.0042 | **0.0000** | 0.0026 | 0.0024 |
| | ESD-x | 0.0064 | 0.0051 | 0.0053 | 0.0060 | 0.0028 | 0.0094 | 0.0058 | 0.0048 | 0.0016 | 0.0005 | **0.0023** | **0.0023** |
| | ESD-u | 0.0081 | 0.0088 | 0.0075 | 0.0077 | 0.0064 | 0.0121 | 0.0084 | 0.0064 | 0.0019 | 0.0021 | 0.0057 | 0.0040 |
| | ErasePro | **0.0007** | **-0.0004** | 0.0047 | 0.0003 | **-0.0001** | **0.0002** | **0.0009** | 0.0049 | **0.0007** | 0.0056 | 0.0039 | 0.0038 |

Table 6: Detailed quantitative results on instance and art style erasure. **Bold** numbers indicate the best performance. Grey numbers denote reference values from the base model (before erasure). A: anchor, T: target, O: other. "–" indicates not applicable.