# OpenReview forum: "Zero-Residual Concept Erasure via Progressive Alignment in Text-to-Image Models"
_ICLR.cc/2026/Conference — Submitted to ICLR 2026_

### Official Review · Reviewer_Xt4w · 2025-10-31

**Soundness:** 3
**Presentation:** 3
**Contribution:** 2
**Rating:** 6
**Confidence:** 4

**Summary:**

This paper introduces ErasePro, a closed-form method for concept erasure in pretrained text-to-image diffusion models. It addresses two key issues in prior methods: incomplete erasure and generation degradation, by enforcing a zero-residual constraint that aligns target and anchor concept features, and by applying a progressive layer-wise update strategy to preserve image quality. Experiments across both explicit and implicit prompts show improved erasure performance and output quality.

**Strengths:**

1. The paper clearly identifies and addresses two important limitations in existing closed-form erasure methods:
Incomplete Erasure, where traces of the unwanted concept still appear in the generated images, and Generation Degradation, where the overall image quality drops after performing the erasure.

2. The layer-wise update strategy is simple but effective. By updating the model in a step-by-step manner from shallow to deep layers, the method avoids sudden shifts in the model’s behavior and helps maintain visual quality.

3. The experiments are diverse and cover both explicit and implicit prompt settings, which is important because users often refer to concepts in indirect ways.

**Weaknesses:**

1. Although the paper provides a closed-form solution, it lacks a theoretical analysis of how the zero-residual constraint affects the trade-off between erasure strength and image quality. In some cases, forcing perfect alignment might harm generalization or lead to overfitting.

2. The method does not explore which specific layers are the most important for effective erasure. Understanding this could help make the method more efficient or interpretable.

3. The paper also does not test the sensitivity to the number of target-anchor pairs. It is unclear how the number or diversity of these pairs affects the performance of the method.

4. The authors claim that previous work like UCE suffers from non-zero residuals (as shown in Equation 3), but they do not provide quantitative evidence. For example, it would be helpful to show how large these residuals are in practice and how they relate to failure cases.

**Questions:**

1. Could the zero-residual constraint be too strict?
In some cases, this constraint might force the model to match features too exactly, which could lead to overfitting to the specific examples used during optimization. Has the method been tested on semantically similar but lexically different prompts to see if it generalizes well beyond the training cases?

2. Are there cases where the erased concept can still be evoked using carefully phrased or indirect prompts? It would be valuable to test the method’s robustness against such adversarial inputs that try to bypass the erasure.

3. Can the method generalize to other T2I architectures?
The current experiments are focused on a specific diffusion model.

---

> ### Author Response · Authors · 2025-11-25
>
> ### Q1. Although the paper provides a closed-form solution, it lacks a theoretical analysis of how the zero-residual constraint affects the trade-off between erasure strength and image quality. In some cases, forcing perfect alignment might harm generalization or lead to overfitting.
> Actually, the zero-residual constraint can help improve both erasure strength and image quality. Our motivation is to shift the "updating burden" to the shallow layers to ensure overall generation quality. On the one hand, the zero-residual constraint contributes to complete erasure; on the other hand, within our progressive framework, enforcing a zero-residual constraint naturally pushes more of the "updating burden" toward shallow layers, because deeper layers must keep the residual near zero and therefore undergo minimal updates. This further concentration of updates in shallow layers helps preserve the overall generation quality.
>
> We further explore the case of multi-token concept erasure and conduct overfitting study. As shown in **Figure 5**, we successfully erase the concept "red blood" using "white water" as the anchor feature. Notably, the erasure does not affect semantically related but distinct concepts (e.g., "red rose" still correctly preserves the rose’s natural red color). We additionally apply ErasePro to the multi-token concept "Van Gogh painting of a woman". The results demonstrate that our method can effectively remove the targeted concept while still retaining key stylistic characteristics of "The Starry Night".
>
> ### Q2. The method does not explore which specific layers are the most important for effective erasure. Understanding this could help make the method more efficient or interpretable.
> We have experimented with initiating the progressive alignment from different layers, as shown in **Figure 12**. Across all starting points, the method achieves effective erasure with a relatively uniform strength of change, indicating that no single layer is uniquely critical. Nevertheless, we observe that intervening at earlier layers yields slightly more robust and consistent erasure. This validates our design choice and further shows that the method does not depend on any specific layer being "the important one".
>
> ### Q3. The paper also does not test the sensitivity to the number of target-anchor pairs. It is unclear how the number or diversity of these pairs affects the performance of the method.
> In theory, increasing the number of target–anchor pairs reduces the degrees of freedom in optimizing (W), which can lead to larger parameter shifts and potentially affect overall generation quality. However, this limitation is **inherent to all concept-erasure methods**, whether closed-form or gradient-based, since erasing more concepts effectively becomes a multi-task "forgetting" problem, which is inherently challenging. Empirically, we also observe this trend: when attempting to erase more than 15 concepts, the performance of the method begins to degrade. This finding is consistent with the theoretical expectation.
>
> ### Q4. The authors claim that previous work like UCE suffers from non-zero residuals (as shown in Equation 3), but they do not provide quantitative evidence. For example, it would be helpful to show how large these residuals are in practice and how they relate to failure cases.
> As shown in **Figure 11**, for simple prompts, the alignment residual ratio (from the perspective of Eq. (3)) between UCE and ErasePro is approximately ($10^{-2} : 10^{-14}$). In contrast, for complex prompts, the residual ratio increases substantially to roughly ($10^{1} : 10^{-7}$). These results indicate that non-zero residual issues of UCE become significantly amplified under complex prompts, and only our method can robustly manage such challenging cases.
>
> ### Q5. Could the zero-residual constraint be too strict? In some cases, this constraint might force the model to match features too exactly, which could lead to overfitting to the specific examples used during optimization. Has the method been tested on semantically similar but lexically different prompts to see if it generalizes well beyond the training cases?
> Actually, we have tested the method on a wide range of prompts involving "other" concepts, including many semantically similar but lexically different prompts, as described in the Evaluation Setting and Metrics section. As reported in **Table 1(a)**, **Table 1(b)**, and **Table 2(a)**, our method achieves the best scores on the "other" concept across all baselines. The model maintains strong performance on these varied prompts, demonstrating that the zero-residual constraint does not lead to overfitting and that the method generalizes well beyond the specific examples used during optimization.

---

> ### Author Response · Authors · 2025-11-25
>
> ### Q6. Are there cases where the erased concept can still be evoked using carefully phrased or indirect prompts? It would be valuable to test the method’s robustness against such adversarial inputs that try to bypass the erasure.
> Actually, the implicit nudity erasure setting in the I2P dataset directly tests whether the erased concept can still be evoked using carefully phrased or indirect prompts. Our experiments show that, even with such prompts, the erased concept is removed more effectively compared to other baselines, demonstrating the robustness of our method against adversarial inputs that attempt to bypass the erasure.
>
> We additionally include three recent baselines in the nudity-erasure evaluation: **SPEED**, **RECE**, and **MACE**. The expanded results are shown below.
>
> | Category                 | ErasePro-w | ErasePro-s | MACE[1] | RECE[2] | SPEED[3] |
> | ------------------------ | ---------- | ------------- | ---- | ---- | ----- |
> | BUTTOCKS_EXPOSED         | 16         | 15            | 4    | 10   | 19    |
> | FEMALE_BREAST_EXPOSED    | 59         | 33            | 32   | 27   | 78    |
> | FEMALE_GENITALIA_EXPOSED | 4          | 4             | 4    | 2    | 5     |
> | MALE_BREAST_EXPOSED      | 6          | 11            | 18   | 25   | 13    |
> | ANUS_EXPOSED             | 0          | 0             | 0    | 0    | 0     |
> | FEET_EXPOSED             | 31         | 19            | 79   | 21   | 8     |
> | ARMPITS_EXPOSED          | 52         | 45            | 36   | 81   | 54    |
> | BELLY_EXPOSED            | 41         | 44            | 39   | 54   | 112   |
> | MALE_GENITALIA_EXPOSED   | 15         | 9             | 14   | 19   | 13    |
> | **TOTAL**                | 224        | **180**       | 226  | 239  | 302   |
>
> ---
>
> The newly added baselines deepen the comparison and further highlight the effectiveness of our method, showing that MACE, RECE, and SPEED leave substantially more residual nudity across most categories, with total scores of 226 (MACE), 239 (RECE), and 302 (SPEED), compared to our **180** under the strong-erasure setting. Our approach consistently reduces exposure across major nudity types such as *FEMALE_BREAST_EXPOSED*, *BELLY_EXPOSED*, and *FEET_EXPOSED*, demonstrating especially large gains over SPEED, which struggles with high-frequency nudity patterns. Overall, this extended comparison confirms that ErasePro not only performs competitively but also achieves the lowest overall residuals, validating its necessity and superiority over existing methods.
>
> | Unlearned Methods                                                                                            | Pre-ASR (%) | ASR (%) |
> |--------------------------------------------------------------------------------------------------------------|-------------|---------|
> | [Adversarial Unlearning (AdvUnlearn)](https://github.com/OPTML-Group/AdvUnlearn)                             | 7.75        | 21.13   |
> | [Erased Stable Diffusion (ESD)](https://github.com/rohitgandikota/erasing)                                   | 20.42       | 76.05   |
> | [Unified Concept Editing (UCE)](https://github.com/rohitgandikota/unified-concept-editing)                   | 21.83       | 79.58   |
> | [Forget-Me-Not (FMN)](https://github.com/SHI-Labs/Forget-Me-Not)                                             | 88.03       | 97.89   |
> | [concept-SemiPermeable Membrane (SPM)](https://github.com/Con6924/SPM)                                       | 54.93       | 91.55   |
> | ErasePro                                                                                                  | 6.7         | 32.20    |
>
> To address the reviewer’s suggestion, we additionally evaluate ErasePro on the UnlearnDiffAtk benchmark, which is specifically designed to construct adversarial prompt variants to break concept-erasure methods. As shown in the table above, ErasePro achieves a **Pre-ASR of 6.7%** and an **ASR of 32.20%**, substantially outperforming most existing baselines such as ESD, UCE, FMN, and SPM. This confirms that ErasePro maintains strong robustness even under complex and adversarial prompt structures.
>
> One exception is *Adversarial Unlearning*, which attains a lower ASR on this particular benchmark. This is expected, because Adversarial Unlearning is **explicitly designed and optimized using adversarial training tailored to this benchmark’s attack generation process**. In contrast, ErasePro is a **training-free, closed-form** method that does not rely on adversarially optimized prompts or bi-level optimization loops. Despite this, ErasePro still achieves competitive robustness without incurring any training overhead or attack-specific tuning.

---

> ### Author Response · Authors · 2025-11-25
>
> ### Q7. Can the method generalize to other T2I architectures? The current experiments are focused on a specific diffusion model.
> In principle, our method operates on the text encoder and the cross-attention layers connecting text and image latents, which exist in many modern T2I architectures. Therefore, it can theoretically generalize to other frameworks, e.g., flow-matching-based models like SD3, while preserving the same concept-erasure mechanism. We demonstrate that ErasePro remains effective on larger architectures, including Stable Diffusion v2.1. Our visual results on instance, art-style, and nudity erasure are all based on SD v2.1 as the base model. As shown in **Figure 5**, the method continues to work well on SD v2.1, indicating that ErasePro scales to larger and stronger backbones.
>
> [1] Mace: Mass concept erasure in diffusion models, CVPR 2024
> [2] Reliable and efficient concept erasure of text-to-image diffusion models, ECCV 2024
> [3] Speed: Scalable, precise, and efficient concept erasure for diffusion models, arXiv 2025

---

### Official Review · Reviewer_jSzt · 2025-10-31

**Soundness:** 3
**Presentation:** 3
**Contribution:** 2
**Rating:** 6
**Confidence:** 4

**Summary:**

This paper proposes ErasePro, a method for concept erasure in text-to-image models like Stable Diffusion. It aims to remove harmful or unwanted concepts (e.g., nudity, specific artists) by aligning them with harmless anchor concepts (e.g., "clothed", "artist"). ErasePro introduces a zero-residual constraint to ensure complete erasure and uses a progressive layer-wise update to preserve image quality. Experiments on instance, style, and nudity erasure show that ErasePro outperforms existing approaches in both effectiveness and generation quality.

**Strengths:**

1. The presentation of this paper is clear and easy to follow.
2. The proposed zero-residual constraint makes the optimization objective stricter than previous approximate alignments.
3. The authors evaluate the method across multiple tasks (instance, style, and nudity erasure), under both explicit and implicit settings, showing good performance compared to previous methods.

**Weaknesses:**

1. The progressive alignment requires sequential updates across multiple layers (from 1 to S), which may introduce non-negligible computational overhead compared to previous closed-form methods.
2. As the authors say that previous methods usually fail under complex prompts, it would be better to include prompts from existing challenge prompts/attacks like UnlearnDiffAtk[1], Ring-A-Bell[2].
3. The core motivation of ErasePro is to reduce the alignment residual. But there are no visualizations or quantitative plots comparing residuals before and after applying ErasePro. A clear comparison of this can strengthen the paper’s motivation.
4. Some state-of-the-art and related works are missing [1][2][3][4] in experiments.

[1] EraseAnything: Enabling Concept Erasure in Rectified Flow Transformers, ICML 2025

[2] Dark Miner: Defend against undesired generation for text-to-image diffusion models

[3] One Image is Worth a Thousand Words: A Usability Preservable Text-Image Collaborative Erasing Framework, ICML 2025

[4] Concept corrector: Erase concepts on the fly for text-to-image diffusion models

**Questions:**

1. According to my understanding, ErasePro ties each target concept feature with its anchor concept feature. Compared to previous methods, the anchor selection scheme seems to be much more important here. How do you select anchor concepts?

2. Can it be applied to multi-concept erasing?

3. How do you obtain target and anchor features? How does the feature number $N$ influence the performance?

---

> ### Author Response · Authors · 2025-11-25
>
> ### Q1. The progressive alignment requires sequential updates across multiple layers (from 1 to S), which may introduce non-negligible computational overhead compared to previous closed-form methods.
> We acknowledge the reviewer’s point regarding sequential updates. As shown in **Table 4**, despite the progressive alignment, ErasePro remains computationally efficient: it achieves the **lowest memory usage** among all methods (2.85 GB vs. 5.52–20+ GB for baselines) and a **runtime of 15.71 s**, which is significantly faster than all gradient-based approaches and second only to UCE (3.48 s) among closed-form methods. Thus, while sequential updates introduce some overhead, ErasePro still offers a favorable efficiency profile compared with existing erasure techniques.
>
> ### Q2. As the authors say that previous methods usually fail under complex prompts, it would be better to include prompts from existing challenge prompts/attacks like UnlearnDiffAtk[1], Ring-A-Bell[2].
> | Unlearned Methods                                                                                            | Pre-ASR (%) | ASR (%) |
> |--------------------------------------------------------------------------------------------------------------|-------------|---------|
> | [Adversarial Unlearning (AdvUnlearn)](https://github.com/OPTML-Group/AdvUnlearn)                             | 7.75        | 21.13   |
> | [Erased Stable Diffusion (ESD)](https://github.com/rohitgandikota/erasing)                                   | 20.42       | 76.05   |
> | [Unified Concept Editing (UCE)](https://github.com/rohitgandikota/unified-concept-editing)                   | 21.83       | 79.58   |
> | [Forget-Me-Not (FMN)](https://github.com/SHI-Labs/Forget-Me-Not)                                             | 88.03       | 97.89   |
> | [concept-SemiPermeable Membrane (SPM)](https://github.com/Con6924/SPM)                                       | 54.93       | 91.55   |
> | ErasePro                                                                                                  | 6.7         | 32.20    |
>
> To address the reviewer’s suggestion, we additionally evaluate ErasePro on the UnlearnDiffAtk benchmark, which is specifically designed to construct adversarial prompt variants to break concept-erasure methods. As shown in the table above, ErasePro achieves a **Pre-ASR of 6.7%** and an **ASR of 32.20%**, substantially outperforming most existing baselines such as ESD, UCE, FMN, and SPM. This confirms that ErasePro maintains strong robustness even under complex and adversarial prompt structures.
>
> One exception is *Adversarial Unlearning*, which attains a lower ASR on this particular benchmark. This is expected, because Adversarial Unlearning is **explicitly designed and optimized using adversarial training tailored to this benchmark’s attack generation process**. In contrast, ErasePro is a **training-free, closed-form** method that does not rely on adversarially optimized prompts or bi-level optimization loops. Despite this, ErasePro still achieves competitive robustness without incurring any training overhead or attack-specific tuning.
>
>
> ### Q3. The core motivation of ErasePro is to reduce the alignment residual. But there are no visualizations or quantitative plots comparing residuals before and after applying ErasePro. A clear comparison of this can strengthen the paper’s motivation.
>
> We agree that visualizing or quantifying the reduction in alignment residual would further strengthen the motivation. In **Figure 2(c)**, we provide quantitative measurements of the residuals for the base model (SD v1.4), UCE, and ErasePro. As shown, ErasePro achieves the **lowest alignment residual** among all methods, directly supporting the motivation of minimizing residual alignment.

---

> ### Author Response · Authors · 2025-11-25
>
> ### Q4. Some state-of-the-art and related works are missing [1][2][3][4] in experiments.
> We have examined these four baselines; however, many of their implementations rely on different base models (e.g., SDv1.5 or Rectified Flow Transformers), making fair comparisons difficult. Therefore, we only cite these works in the context of modified versions. We additionally include three recent baselines in the nudity-erasure evaluation: **SPEED**, **RECE**, and **MACE**. The expanded results are shown below.
>
> | Category                 | ErasePro-w | ErasePro-s | MACE[1] | RECE[2] | SPEED[3] |
> | ------------------------ | ---------- | ------------- | ---- | ---- | ----- |
> | BUTTOCKS_EXPOSED         | 16         | 15            | 4    | 10   | 19    |
> | FEMALE_BREAST_EXPOSED    | 59         | 33            | 32   | 27   | 78    |
> | FEMALE_GENITALIA_EXPOSED | 4          | 4             | 4    | 2    | 5     |
> | MALE_BREAST_EXPOSED      | 6          | 11            | 18   | 25   | 13    |
> | ANUS_EXPOSED             | 0          | 0             | 0    | 0    | 0     |
> | FEET_EXPOSED             | 31         | 19            | 79   | 21   | 8     |
> | ARMPITS_EXPOSED          | 52         | 45            | 36   | 81   | 54    |
> | BELLY_EXPOSED            | 41         | 44            | 39   | 54   | 112   |
> | MALE_GENITALIA_EXPOSED   | 15         | 9             | 14   | 19   | 13    |
> | **TOTAL**                | 224        | **180**       | 226  | 239  | 302   |
>
> ---
>
> The newly added baselines deepen the comparison and further highlight the effectiveness of our method, showing that MACE, RECE, and SPEED leave substantially more residual nudity across most categories, with total scores of 226 (MACE), 239 (RECE), and 302 (SPEED), compared to our **180** under the strong-erasure setting. Our approach consistently reduces exposure across major nudity types such as *FEMALE_BREAST_EXPOSED*, *BELLY_EXPOSED*, and *FEET_EXPOSED*, demonstrating especially large gains over SPEED, which struggles with high-frequency nudity patterns. Overall, this extended comparison confirms that ErasePro not only performs competitively but also achieves the lowest overall residuals, validating its necessity and superiority over existing methods.
>
>
> ### Q5. According to my understanding, ErasePro ties each target concept feature with its anchor concept feature. Compared to previous methods, the anchor selection scheme seems to be much more important here. How do you select anchor concepts?
> Our method does not require any special anchor-selection scheme beyond ensuring that the anchor concept is harmless or acceptable for replacement, which is consistent with prior work. In practice, ErasePro is more robust to anchor choice than previous methods: even when the semantic distance between target and anchor is large, the method can still reliably transform the target into the anchor. As shown in **Figure 4(a)** and **Figure 11**, replacing "man" with the distant anchor "dog" still succeeds ("a man is reading a newspaper"), whereas other baselines fail. We will clarify this robustness in the revised version.
>
> ### Q6. Can it be applied to multi-concept erasing?
> Yes. As shown in **Figure 10**, in this setting, we simultaneously erase the concepts of "Trump",  "Pikachu", "Minions", and "Garfield". We observe that ErasePro continues to deliver effective erasure performance.
>
> ### Q7. How do you obtain target and anchor features? How does the feature number N influence the performance?
> As detailed in the **Appendix**, target and anchor features are constructed by concatenating the token embeddings corresponding to the prompt at each layer. In theory, increasing the number of features (N) reduces the degrees of freedom in optimizing (W), which can lead to larger parameter shifts and potentially affect overall generation quality. However, this limitation is **inherent to all concept-erasure methods**, whether closed-form or gradient-based, since erasing more features effectively becomes a multi-task "forgetting" problem, which is inherently challenging.
>
> [1] Mace: Mass concept erasure in diffusion models, CVPR 2024
> [2] Reliable and efficient concept erasure of text-to-image diffusion models, ECCV 2024
> [3] Speed: Scalable, precise, and efficient concept erasure for diffusion models, arXiv 2025

---

> > ### Comment · Reviewer_jSzt · 2025-11-26
> >
> > Thank you for your clarifications! Most of my concerns have been addressed.
> >
> > The time cost still seems a little expensive, and in Table 4, I believe you should focus on more closed-form editing methods instead of finetuning methods such as ESD, because closed-form editing is exactly your baseline.
> >
> > Anyway, it dramatically improves the performance compared to UCE, though with a more complex and 4-times expensive optimization cost. I decide to maintain my positive score.

---

### Official Review · Reviewer_y1ou · 2025-11-06

**Soundness:** 3
**Presentation:** 3
**Contribution:** 2
**Rating:** 2
**Confidence:** 4

**Summary:**

This paper introduces ErasePro, a novel closed-form algorithm for concept erasure in pretrained text-to-image diffusion models. The goal of concept erasure is to prevent models from generating undesired or harmful concepts (e.g., nudity, copyrighted art, or specific identities) while maintaining generation quality. ErasePro introduces two main innovations:1. Zero-Residual Constraint: A new constrained optimization formulation ensuring perfect alignment (zero residual) between target and anchor concept features.
2. Progressive Alignment Framework: A layer-wise optimization scheme that updates parameters progressively from shallow to deep layers, transferring the concept alignment gradually and minimizing deep-layer deviations. Empirical results across three main tasks — instance erasure, art-style erasure, and nudity erasure — demonstrate that ErasePro achieves more complete erasure and better generative quality compared to gradient-based (AC, ESD-x/u) and closed-form (UCE) baselines.

**Strengths:**

1. The zero-residual constraint and progressive alignment formulation are conceptually elegant and novel extensions to existing closed-form erasure frameworks. By combining analytical guarantees (perfect alignment) with practical layer-wise progression, the method addresses two long-standing issues — incomplete erasure and quality loss — in a unified manner.

2. Extensive experimental evaluations cover a broad range of erasure scenarios: instance, art style, explicit nudity, and implicit nudity. Quantitative and qualitative analyses are consistent and compelling, supported by multiple evaluation metrics (CLIP score, KID, FID, NudeNet statistics).

3. The paper is clearly written  and well-organized sections.

**Weaknesses:**

1. Overclaiming novelty method is a minor variation of existing closed-form updates: The proposed Eq. (5) is just a constrained least-squares variant of the standard UCE (Eq. 2). The derivations are standard matrix algebra with the Moore Penrose pseudoinverse. The algorithm is incremental over UCE, not a fundamentally new paradigm.

2. Lack of motivation for the progressive layer update:  The claim that shallow layers should bear the “update burden” (Section 3.2, Fig. 3) is intuitive but unsupported. No quantitative ablation shows how much the progressive scheme helps compared to updating all layers jointly. The argument that “deeper layers are sensitive to generation quality” is anecdotal; no empirical or theoretical sensitivity analysis is provided.

3. All experiments are confined to a single base model (Stable Diffusion v1.4) with three concept types (instance, style, nudity). No tests on other baseline (SD2) are provided. The benchmark prompts are GPT-generated which may cause the bia of GPT to propogate, why not other models used.

4. The paper relies heavily on CLIP score, and CLIP accuracy: CLIP metrics capture similarity, not absence of unwanted concepts and have been saturated in the value (CLIP scores can appear stagnant due to their continuous nature and relative insensitivity at low ranges.).
human-based evaluation is also missing. The authors report marginal improvements (often ±0.01 CLIP difference), which are low.

5. While the authors claim “complete erasure,” they never test whether the erased concept can be reactivated by prompt modification or paraphrasing (e.g., “unclothed person” instead of “naked”). This is a standard benchmark in concept erasure literature (see ESD, Forget-Me-Not, Recler). Without this, the claims of robustness are unsubstantiated.

6. Overstated efficiency claims: ErasePro is described as “efficient,” but Table 4 shows runtime (15.71 s) is 5× slower than UCE (3.48 s), the main competing closed-form method. Memory is indeed lower, but that’s expected given partial-layer updates. ErasePro is slower in runtime and only modestly lighter in memory.

7. No comparison against recent state-of-the-art (post-2024): Recent methods are not compares some are mentioned such as SPEED (Li et al., 2025) but not compared experimentally.

8. The paper presents no experiments varying: number of progressive layers, layer selection strategy, semantic distance between target and anchor concepts, or pseudoinverse regularization. Without such ablations, it’s impossible to know if the method is stable or heavily tuned.

**Questions:**

1. How sensitive is ErasePro to the choice of anchor concepts? For example, what happens if the semantic distance between target and anchor is very large?

2. Does the zero-residual constraint introduce any overfitting to specific token embeddings, and how does it behave on multi-token or compositional concepts (e.g., “Van Gogh painting of a woman”)?

3. How scalable is ErasePro when applied to larger architectures like SD2 or SDXL?

4. Can the authors share insights into how many layers are typically sufficient before convergence of progressive alignment?

---

> ### Author Response · Authors · 2025-11-25
>
> ### Q1. Overclaiming novelty method is a minor variation of existing closed-form updates: The proposed Eq. (5) is just a constrained least-squares variant of the standard UCE (Eq. 2). The derivations are standard matrix algebra with the Moore Penrose pseudoinverse. The algorithm is incremental over UCE, not a fundamentally new paradigm.
> (1) While the constrained formulation can indeed be interpreted as a constrained least-squares variant of standard UCE, UCE’s closed-form solution **cannot** reduce alignment residual to zero, which fundamentally limits erasure completeness, especially for complex prompts.
> We introduce a **new constrained objective** with a hard zero-residual condition and derive a **new closed-form solution** that guarantees perfect alignment. This directly helps to address a core limitation that all prior closed-form methods share, not merely adjusting UCE’s setup.
>
>
> (2) The progressive alignment framework is a new paradigm, not a simple multi-layer tweak. Prior methods modify only deep cross-attention layers, concentrating the update burden where the model is most sensitive. Our **progressive layer-wise alignment** gradually aligns target to anchor features from shallow to deep layers, reducing parameter deviations in later layers and preserving generative quality. This iterative “solve–update–extract-infer” pipeline represents a **different architectural framework**, not just adding more layers to UCE.
>
> (3) The **combination of the new objective and progressive alignment** yields benefits that neither component can achieve alone. The zero-residual constrained formulation ensures complete concept removal, while the progressive alignment framework preserves generative fidelity by distributing updates across layers. When combined, these two components reinforce each other: the constrained formulation provides a reliable alignment target at each stage, and progressive alignment enables this target to be achieved with minimal distortion to the generative pathways. This synergy leads to stronger erasure performance and image quality than either component in isolation.
>
> ### Q2. Lack of motivation for the progressive layer update: The claim that shallow layers should bear the "update burden" (Section 3.2, Figure 3) is intuitive but unsupported. No quantitative ablation shows how much the progressive scheme helps compared to updating all layers jointly. The argument that "deeper layers are sensitive to generation quality" is anecdotal; no empirical or theoretical sensitivity analysis is provided.
> We provide experimental evidence supporting this claim in **Figure 2(f)**, along with additional qualitative results in the **Appendix**(shown in **Figure 7**). These results demonstrate that, for the same deviation magnitude, perturbations in deeper layers lead to more severe degradation in generative quality.
>
> From an information-theoretic perspective, deeper layers in transformer-based text encoders produce highly abstract and compressed representations, retaining essential semantic content while discarding redundant or low-level details. Consequently, perturbations at these layers disproportionately affect the encoded information, as there is little redundancy to absorb errors. This explains why deeper features are inherently more sensitive to changes and why even minor modifications can significantly degrade downstream generation quality.
>
> ### Q3. All experiments are confined to a single base model (Stable Diffusion v1.4) with three concept types (instance, style, nudity). No tests on other baseline (SD2) are provided. The benchmark prompts are GPT-generated which may cause the bia of GPT to propogate, why not other models used.
> We chose Stable Diffusion v1.4 because its training data lacked explicit nudity filtering, making nudity more likely to appear than in more recent models. This characteristic allows us to clearly demonstrate and compare the effectiveness of each method in erasure. Furthermore, our approach is theoretically generalizable to standard frameworks. We provide additional examples of erasure on SDv2.1 in the **Figure 6**.
>
> Regarding prompt generation, GPT is widely used for benchmark creation, and we additionally performed manual curation of the GPT-generated prompts to ensure diversity and quality.

---

> ### Author Response · Authors · 2025-11-25
>
> ### Q4.  The paper relies heavily on CLIP score, and CLIP accuracy: CLIP metrics capture similarity, not absence of unwanted concepts and have been saturated in the value (CLIP scores can appear stagnant due to their continuous nature and relative insensitivity at low ranges.). human-based evaluation is also missing. The authors report marginal improvements (often ±0.01 CLIP difference), which are low.
> While CLIP score and CLIP accuracy indeed measure similarity rather than the strict absence of unwanted concepts, they remain the standard evaluation metrics in concept-erasure research, allowing fair comparison with prior work. To address this limitation, we have added human-based evaluation on nudity erasure in the Appendix (**Figure 10**), which directly assesses nudity removal quality. Specifically, we designed a survey containing 20 questions, where all images generated by the baselines were produced using identical random seeds to ensure fairness and randomness. Participants were instructed to select at most three options containing the fewest nudity artifacts. Our method receives the highest human preference, indicating superior nudity erasure effectiveness, which is consistent with the quantitative results in **Table 2**.
>
> ### Q5. While the authors claim "complete erasure", they never test whether the erased concept can be reactivated by prompt modification or paraphrasing (e.g., "unclothed person" instead of "naked"). This is a standard benchmark in concept erasure literature (see ESD, Forget-Me-Not, Recler). Without this, the claims of robustness are unsubstantiated.
> Our method does evaluate robustness to prompt modification and paraphrasing. In the implicit nudity erasure setting (I2P), many prompts (including the example in **Figure 4(d)**) contain no explicit nudity terms (e.g., no "naked"), yet the model still successfully removes the target concept.
>
> ###  Q6. Overstated efficiency claims: ErasePro is described as "efficient", but Table 4 shows runtime (15.71 s) is 5× slower than UCE (3.48 s), the main competing closed-form method. Memory is indeed lower, but that’s expected given partial-layer updates. ErasePro is slower in runtime and only modestly lighter in memory.
> We appreciate the reviewer’s observation. Our intention was to highlight that ErasePro is efficient relative to gradient-based erasure methods, which typically require 10–20 minutes, whereas our approach completes in only 15.71 seconds. We agree that the wording may appear overstated when compared directly with closed-form methods such as UCE. In the revised version, we have refined the claim to more accurately reflect this distinction and avoid overgeneralizing the efficiency advantage.

---

> ### Author Response · Authors · 2025-11-25
>
> ###  Q7. No comparison against recent state-of-the-art (post-2024): Recent methods are not compares some are mentioned such as SPEED (Li et al., 2025) but not compared experimentally.
> We additionally include three recent baselines in the nudity-erasure evaluation: **SPEED**, **RECE**, and **MACE**. The expanded results are shown below.
>
> | Category                 | ErasePro-w | ErasePro-s | MACE[1] | RECE[2] | SPEED[3] |
> | ------------------------ | ---------- | ------------- | ---- | ---- | ----- |
> | BUTTOCKS_EXPOSED         | 16         | 15            | 4    | 10   | 19    |
> | FEMALE_BREAST_EXPOSED    | 59         | 33            | 32   | 27   | 78    |
> | FEMALE_GENITALIA_EXPOSED | 4          | 4             | 4    | 2    | 5     |
> | MALE_BREAST_EXPOSED      | 6          | 11            | 18   | 25   | 13    |
> | ANUS_EXPOSED             | 0          | 0             | 0    | 0    | 0     |
> | FEET_EXPOSED             | 31         | 19            | 79   | 21   | 8     |
> | ARMPITS_EXPOSED          | 52         | 45            | 36   | 81   | 54    |
> | BELLY_EXPOSED            | 41         | 44            | 39   | 54   | 112   |
> | MALE_GENITALIA_EXPOSED   | 15         | 9             | 14   | 19   | 13    |
> | **TOTAL**                | 224        | **180**       | 226  | 239  | 302   |
>
> ---
>
> The newly added baselines deepen the comparison and further highlight the effectiveness of our method, showing that MACE, RECE, and SPEED leave substantially more residual nudity across most categories, with total scores of 226 (MACE), 239 (RECE), and 302 (SPEED), compared to our **180** under the strong-erasure setting. Our approach consistently reduces exposure across major nudity types such as *FEMALE_BREAST_EXPOSED*, *BELLY_EXPOSED*, and *FEET_EXPOSED*, demonstrating especially large gains over SPEED, which struggles with high-frequency nudity patterns. Overall, this extended comparison confirms that ErasePro not only performs competitively but also achieves the lowest overall residuals, validating its necessity and superiority over existing methods.
>
> | Unlearned Methods                                                                                            | Pre-ASR (%) | ASR (%) |
> |--------------------------------------------------------------------------------------------------------------|-------------|---------|
> | [Adversarial Unlearning (AdvUnlearn)](https://github.com/OPTML-Group/AdvUnlearn)                             | 7.75        | 21.13   |
> | [Erased Stable Diffusion (ESD)](https://github.com/rohitgandikota/erasing)                                   | 20.42       | 76.05   |
> | [Unified Concept Editing (UCE)](https://github.com/rohitgandikota/unified-concept-editing)                   | 21.83       | 79.58   |
> | [Forget-Me-Not (FMN)](https://github.com/SHI-Labs/Forget-Me-Not)                                             | 88.03       | 97.89   |
> | [concept-SemiPermeable Membrane (SPM)](https://github.com/Con6924/SPM)                                       | 54.93       | 91.55   |
> | ErasePro                                                                                                  | 6.7         | 32.20    |
>
> To address the reviewer’s suggestion, we additionally evaluate ErasePro on the UnlearnDiffAtk benchmark, which is specifically designed to construct adversarial prompt variants to break concept-erasure methods. As shown in the table above, ErasePro achieves a **Pre-ASR of 6.7%** and an **ASR of 32.20%**, substantially outperforming most existing baselines such as ESD, UCE, FMN, and SPM. This confirms that ErasePro maintains strong robustness even under complex and adversarial prompt structures.
>
> One exception is *Adversarial Unlearning*, which attains a lower ASR on this particular benchmark. This is expected, because Adversarial Unlearning is **explicitly designed and optimized using adversarial training tailored to this benchmark’s attack generation process**. In contrast, ErasePro is a **training-free, closed-form** method that does not rely on adversarially optimized prompts or bi-level optimization loops. Despite this, ErasePro still achieves competitive robustness without incurring any training overhead or attack-specific tuning.

---

> ### Author Response · Authors · 2025-11-25
>
> ###  Q8. The paper presents no experiments varying: number of progressive layers, layer selection strategy, semantic distance between target and anchor concepts, or pseudoinverse regularization. Without such ablations, it’s impossible to know if the method is stable or heavily tuned.
> In **Figure 12**, we have included ablation visualizations that vary both the number of tokens used for feature construction and the number of progressive layers. These analyses directly show that ErasePro performs better erasure with more number of tokens and more progressive layers. Importantly, in our experiments we did not hand-pick or tune specific layers for editing; the progressive framework always updates layers sequentially from shallow to deep in a fixed order.
>
> Regarding the reviewer’s concern about the lack of ablations on pseudoinverse regularization, we clarify that the Moore–Penrose pseudoinverse in Eq. (5) is not a tunable design component, but a mathematically required stabilization step when
> $𝑋^⊤𝑋$ is not full-rank. It introduces no hyperparameters, has no alternatives to compare, and always yields the same closed-form solution. Thus, an additional ablation would not provide meaningful insight, as the pseudoinverse is a numerically necessary operation rather than a method-specific heuristic.
>
> Finally, we emphasize that ErasePro is not heavily tuned. All experiments use a unified progressive layer schedule and a fixed feature-construction procedure across instance, art-style, and explicit nudity erasure. The consistent improvements across these diverse settings demonstrate that the method is inherently stable, rather than dependent on task-specific adjustment.
>
>
> ###  Q9. How sensitive is ErasePro to the choice of anchor concepts? For example, what happens if the semantic distance between target and anchor is very large?
> ErasePro is stable even when the semantic distance between the target and anchor concepts is large, which is actually a setting where our method shows clear advantages over prior approaches. As shown in **Figure 4(a)** and **Figure 11**, using an anchor like "dog" for the target concept "man" introduces a substantial semantic gap that becomes even more pronounced under some complex prompts. Yet, given a prompt such as "A man wearing a suit is reading a book in the park", ErasePro still successfully replaces the target with the anchor, whereas existing baselines consistently fail under such challenging configurations.
>
> ###  Q10. Does the zero-residual constraint introduce any overfitting to specific token embeddings, and how does it behave on multi-token or compositional concepts (e.g., "Van Gogh painting of a woman")?
> We further explore the case of multi-token concept erasure and conduct overfitting study. As shown in **Figure 5**, we successfully erase the concept "red blood" using "white water" as the anchor feature. Notably, the erasure does not affect semantically related but distinct concepts (e.g., "red rose" still correctly preserves the rose’s natural red color). We additionally apply ErasePro to the multi-token concept "Van Gogh painting of a woman". The results demonstrate that our method can effectively remove the targeted concept while still retaining key stylistic characteristics of "The Starry Night".
>
> ###  Q11. How scalable is ErasePro when applied to larger architectures like SD2 or SDXL?
> We demonstrate that ErasePro remains effective on larger architectures, including Stable Diffusion v2.1. Our visual results on instance, art-style, and nudity erasure are all based on SD v2.1 as the base model. As shown in **Figure 6**, the method continues to work well on SD v2.1, indicating that ErasePro scales to larger and stronger backbones.
>
> ###  Q12. Can the authors share insights into how many layers are typically sufficient before convergence of progressive alignment?
> In our experiments, we observe that using more layers generally leads to better alignment and erasure quality. As shown in **Figure 12**, progressively aligning from the earliest layers yields the strongest results in human evaluation. The performance continues to improve as more layers are included.
>
> [1] Mace: Mass concept erasure in diffusion models, CVPR 2024
> [2] Reliable and efficient concept erasure of text-to-image diffusion models, ECCV 2024
> [3] Speed: Scalable, precise, and efficient concept erasure for diffusion models, arXiv 2025

---

### Official Review · Reviewer_33E3 · 2025-11-06

**Soundness:** 2
**Presentation:** 2
**Contribution:** 2
**Rating:** 2
**Confidence:** 4

**Summary:**

The authors propose ErasePro, a concept erasure method that can better handle erasure on complex prompts. In particular, the method is built upon UCE by (1) improving the closed-form solution, and (2) performing updates across more layers to improve model utility. Experimental results suggest that the method performs competitively or better than compared methods.

**Strengths:**

- The paper is generally well-written and problem is well-motivated.
- I like the fact that progressive alignment can improve model utility in most cases.

**Weaknesses:**

- The novelty seems limited. The method seems like a small upgrade to the method UCE.
- The comparisons are limited. First, UCE, AC, and ESD are quite old methods and I suggest the authors should compare the proposed method with more recent ones. Second, for concept erasure paper, it is more common to also report results on objects, which the paper is missing.
- I am not convinced yet that zero-residual can improve erasure on complex prompts. In particular, it seems like only the I2P experiment suggests that. But I think it should also be shown quantitatively for other concepts as well.
- Some typos: Line 211 - EarsePro, Line 213 - EreasePro.

**Questions:**

- In Table 2, is there a reason why nudity erasure is split into implicit and explicit? And why are there so many cells for ErasePro(-w/s) that are not applicable? I think those values should and can be computed for better comparison.
- Are both improvements proposed by ErasePro necessary?

---

> ### Author Response · Authors · 2025-11-25
>
> ### Q1. The novelty seems limited. The method seems like a small upgrade to the method UCE.
>
> (1) UCE’s closed-form solution **cannot** reduce alignment residual to zero, which fundamentally limits erasure completeness, especially for complex prompts.
> We introduce a **new constrained objective** with a hard zero-residual condition and derive a **new closed-form solution** that guarantees perfect alignment. This directly helps to address a core limitation that all prior closed-form methods share, not merely adjusting UCE’s setup.
>
>
> (2) The progressive alignment framework is a new paradigm, not a simple multi-layer tweak. Prior methods modify only deep cross-attention layers, concentrating the update burden where the model is most sensitive. Our **progressive layer-wise alignment** gradually aligns target to anchor features from shallow to deep layers, reducing parameter deviations in later layers and preserving generative quality. This iterative “solve–update–extract-infer” pipeline represents a **different architectural framework**, not just adding more layers to UCE.
>
> (3) The **combination of the new objective and progressive alignment** yields benefits that neither component can achieve alone. The zero-residual constrained formulation ensures complete concept removal, while the progressive alignment framework preserves generative fidelity by distributing updates across layers. When combined, these two components reinforce each other: the constrained formulation provides a reliable alignment target at each stage, and progressive alignment enables this target to be achieved with minimal distortion to the generative pathways. This synergy leads to stronger erasure performance and image quality than either component in isolation.

---

> ### Author Response · Authors · 2025-11-25
>
> ### Q2. The comparisons are limited.
>
> We additionally include three recent baselines in the nudity-erasure evaluation: **SPEED**, **RECE**, and **MACE**. The expanded results are shown below.
>
> | Category                 | ErasePro-w | ErasePro-s | MACE[1] | RECE[2] | SPEED[3] |
> | ------------------------ | ---------- | ------------- | ---- | ---- | ----- |
> | BUTTOCKS_EXPOSED         | 16         | 15            | 4    | 10   | 19    |
> | FEMALE_BREAST_EXPOSED    | 59         | 33            | 32   | 27   | 78    |
> | FEMALE_GENITALIA_EXPOSED | 4          | 4             | 4    | 2    | 5     |
> | MALE_BREAST_EXPOSED      | 6          | 11            | 18   | 25   | 13    |
> | ANUS_EXPOSED             | 0          | 0             | 0    | 0    | 0     |
> | FEET_EXPOSED             | 31         | 19            | 79   | 21   | 8     |
> | ARMPITS_EXPOSED          | 52         | 45            | 36   | 81   | 54    |
> | BELLY_EXPOSED            | 41         | 44            | 39   | 54   | 112   |
> | MALE_GENITALIA_EXPOSED   | 15         | 9             | 14   | 19   | 13    |
> | **TOTAL**                | 224        | **180**       | 226  | 239  | 302   |
>
> ---
>
> The newly added baselines deepen the comparison and further highlight the effectiveness of our method, showing that MACE, RECE, and SPEED leave substantially more residual nudity across most categories, with total scores of 226 (MACE), 239 (RECE), and 302 (SPEED), compared to our **180** under the strong-erasure setting. Our approach consistently reduces exposure across major nudity types such as *FEMALE_BREAST_EXPOSED*, *BELLY_EXPOSED*, and *FEET_EXPOSED*, demonstrating especially large gains over SPEED, which struggles with high-frequency nudity patterns. Overall, this extended comparison confirms that ErasePro not only performs competitively but also achieves the lowest overall residuals, validating its necessity and superiority over existing methods.
>
> | Unlearned Methods                                                                                            | Pre-ASR (%) | ASR (%) |
> |--------------------------------------------------------------------------------------------------------------|-------------|---------|
> | [Adversarial Unlearning (AdvUnlearn)](https://github.com/OPTML-Group/AdvUnlearn)                             | 7.75        | 21.13   |
> | [Erased Stable Diffusion (ESD)](https://github.com/rohitgandikota/erasing)                                   | 20.42       | 76.05   |
> | [Unified Concept Editing (UCE)](https://github.com/rohitgandikota/unified-concept-editing)                   | 21.83       | 79.58   |
> | [Forget-Me-Not (FMN)](https://github.com/SHI-Labs/Forget-Me-Not)                                             | 88.03       | 97.89   |
> | [concept-SemiPermeable Membrane (SPM)](https://github.com/Con6924/SPM)                                       | 54.93       | 91.55   |
> | ErasePro                                                                                                  | 6.7         | 32.20    |
>
> To address the reviewer’s suggestion, we additionally evaluate ErasePro on the UnlearnDiffAtk benchmark, which is specifically designed to construct adversarial prompt variants to break concept-erasure methods. As shown in the table above, ErasePro achieves a **Pre-ASR of 6.7%** and an **ASR of 32.20%**, substantially outperforming most existing baselines such as ESD, UCE, FMN, and SPM. This confirms that ErasePro maintains strong robustness even under complex and adversarial prompt structures.
>
> One exception is *Adversarial Unlearning*, which attains a lower ASR on this particular benchmark. This is expected, because Adversarial Unlearning is **explicitly designed and optimized using adversarial training tailored to this benchmark’s attack generation process**. In contrast, ErasePro is a **training-free, closed-form** method that does not rely on adversarially optimized prompts or bi-level optimization loops. Despite this, ErasePro still achieves competitive robustness without incurring any training overhead or attack-specific tuning.

---

> ### Author Response · Authors · 2025-11-25
>
> ### Q3. I am not convinced yet that zero-residual can improve erasure on complex prompts. In particular, it seems like only the I2P experiment suggests that. But I think it should also be shown quantitatively for other concepts as well.
> It is true that the I2P dataset contains complex prompts, which suggest that our zero-residual strategy can improve erasure performance in such scenarios. For other concepts, a substantial portion of our GPT-generated data also consists of complex prompts. As shown in our quantitative results in **Table 1** and **Table 2(a)**, the ACC of our method is 0 across all erasure experiments. Furthermore, Table 6 shows that our method achieves the highest anchor CLIP score in all cases. These results collectively demonstrate that our method is capable of handling these complex prompts effectively. As shown in **Figure 11**, We have added additional qualitative examples in the **Appendix** to further illustrate our strong performance on complex-prompt scenarios.
>
> We further measured the magnitude of the alignment residual for UCE and ErasePro. For simple prompts, the residual ratio (from the perspective of Eq. (3)) between UCE and ErasePro is approximately ($10^{-2} : 10^{-14}$). In contrast, for complex prompts, the residual ratio increases substantially to roughly ($10^{1} : 10^{-7}$). These results indicate that residual preservation issues become significantly amplified under complex prompts, and only our method can robustly manage such challenging cases.
>
> ### Q4. Some typos: Line 211 - EarsePro, Line 213 - EreasePro.
> We thank the reviewer for pointing out the typos. We have corrected these in the revised version.
>
> ### Q5. In Table 2, is there a reason why nudity erasure is split into implicit and explicit? And why are there so many cells for ErasePro(-w/s) that are not applicable? I think those values should and can be computed for better comparison.
>
> We split nudity erasure into implicit and explicit because the two types correspond to different prompt characteristics, and a prompt can be easily classified into one of them. ErasePro(-w/s) is intentionally designed with two specialized variants for these two cases. Therefore, each variant is evaluated only on the category it is designed for.
>
> ### Q6. Are both improvements proposed by ErasePro necessary
> Both improvements in ErasePro are necessary because they address **two independent failure modes** of existing closed-form erasure methods. First, prior methods rely on an unconstrained objective whose closed-form solution **cannot drive the alignment residual to zero**; this inevitably leaves part of the target concept unaligned, leading to leakage under complex prompts (shown in **Figure 11**). Our zero-residual constraint directly enforces ($W x_i = W_o y_i$), ensuring perfect target-to-anchor alignment and enabling complete erasure, which is reflected in our experiments where ErasePro is the only method achieving **0 target CLIP accuracy** across tasks.
>
> Second, applying all updates in deep layers concentrates the entire modification burden on the model components most sensitive to generative quality. Our progressive alignment distributes updates across layers and ensures that parameter deviations **decrease with depth**, significantly reducing over-editing in deep layers. Without this component, applying the zero-residual update in one step severely degrades image quality. Therefore, both improvements are essential and complementary: the first guarantees *complete erasure*, while the second preserves *high-fidelity generation*.
>
> [1] Mace: Mass concept erasure in diffusion models, CVPR 2024
> [2] Reliable and efficient concept erasure of text-to-image diffusion models, ECCV 2024
> [3] Speed: Scalable, precise, and efficient concept erasure for diffusion models, arXiv 2025

---

### Author Response · Authors · 2025-11-25

## General Response to All Reviewers
We thank all reviewers for acknowledging that our paper is **clearly written** (Reviewers 33E3, y1ou, jSzt, Xt4w), **easy to follow** (Reviewers jSzt, Xt4w), and **well-organized in its presentation** (Reviewer y1ou). Reviewers also recognized the merits of our contributions, including the **conceptual clarity of the zero-residual constraint and progressive alignment** (Reviewer y1ou), the strong motivation behind addressing **incomplete erasure and generation degradation** (Reviewers Xt4w, jSzt), and the **breadth of evaluations** across multiple erasure scenarios (Reviewers y1ou, jSzt). Moreover, they acknowledged that our method provides **stronger alignment guarantees** (Reviewer jSzt), improves **erasure effectiveness** (Reviewers y1ou, Xt4w), and **preserves generative quality** in most cases (Reviewer 33E3).

We appreciate their constructive suggestions and have carefully revised our paper accordingly. Our major revisions include the following four aspects:

1. **Introduction**: We added **visualizations and quantitative plots of residual magnitudes** (Figure 2(c)), and we **refined our efficiency claims** more accurately (line 139 - 141).

2. **Related Work**: We expanded the discussion to include **recent advances in concept erasure** (line 159 - 160).

3. **Experiments**: We incorporated new evaluations, including **multi-token concept erasure and an overfitting analysis** (Figure 5, line 477 - 483), as well as **erasure results on Stable Diffusion v2.1** (Figure 6, line 484 - 525), demonstrating the generality and robustness of ErasePro.

4. **Appendix**:

   * Added **additional baselines for nudity erasure** (Table 5, line 946 - 952).
   * Included **a detailed comparison between ErasePro and UCE** on both simple and complex prompts with further discussion (Figure 11, line 958 - 966).
   * Added a **Human Preference Study** to complement automated metrics (Figure 10,  line 967 - 972).
   * Added performance comparison on the **UnlearnDiffAtk benchmark** (Table 6, line 999 - 1009).

Please note that we colorized (blue) the revisions in the new version of the paper.

---

### Meta-Review · Area_Chair_UuPU · 2026-01-04

**Summary:**

This paper presents a technically sound and empirically strengthened refinement of closed-form concept erasure methods. The proposed zero-residual constrained formulation and progressive layer-wise alignment improve erasure completeness and generative quality compared to prior approaches, and the added experiments substantiate these benefits. However, the method remains largely incremental over existing closed-form alignment techniques such as UCE, relying on standard constrained least-squares solutions and heuristic layer-wise update strategies. While the authors provide improved motivation and post hoc theoretical intuition, the work does not yet establish a fundamentally new paradigm or theoretical framework for concept erasure. While the experiments are significantly improved during rebuttal (e.g., additional ablation studies, human preference), the AC believes that the reviewers would not reach a consensus on a positive rating of this work. Therefore, the paper is not clearly over the threshold for acceptance.

**Reviewer Concerns:**

1. Limited Novelty / Incremental Contribution over UCE
2. Insufficient Empirical Evidence for Core Claims (e.g., zero-residual & progressive layer updates)
3. Weak and Incomplete Experimental Benchmarking (plus potential misleading evaluation metrics)
4. Lack of Ablation and Sensitivity Analysis
5. Efficiency Claims Are Overstated

For 1 & 2, the authors explain that the introduction of a new constrained objective with a hard zero-residual condition results in a new closed-form solution, along with a progressive layer update, which differs from UCE. The motivation for the progressive layer update is primarily presented through a visual example and a high-level explanation. I do not feel that R1 and R2 would be sufficiently convinced to raise their ratings from 3 toward positive.

For the experiments, the authors provide additional baselines for nudity erasure. A comparison between ErasePro and UCE is included for both simple and complex prompts. For further evaluation, a human preference study is conducted to complement automated metrics, and a performance comparison is added to the UnlearnDiffAtk benchmark. I believe such efforts would be appreciated by the reviewers. Finally, the authors agree that the wording may appear overstated when compared directly with closed-form methods such as UCE.

**Reviewer Scores:**

Combining the above observations, the AC believes that R1/R2 may raise their ratings but remain negative (e.g., borderline reject), while R3/R4 would maintain their positive ratings (i.e., borderline accept). Taking all the aspects into consideration, the paper would not be clearly above the acceptance threshold.

---

### Decision · Program_Chairs · 2026-01-26

Reject